# DRef: A Benchmark with Diverse Referring Expressions for Object Comprehension of Vision-Language Models

## Abstract

Referring expression comprehension (REC) tasks challenge vision-language models (VLMs) to locate specific objects within images based on natural-language descriptions, typically by generating bounding boxes or segmentation masks. Existing REC benchmarks suffer from fundamental shortcomings: (1) their limited diversity of referring expressions per object makes it impossible to distinguish whether VLMs truly understand object semantics or simply memorize specific associations; (2) the evaluation metrics do not reveal whether a VLM is robust enough to face complex and diverse referring expressions. We address these issues with a novel benchmark and two innovative metrics. Our benchmark, **D**iverse **Ref**erring Expressions for Object Comprehension (DRef), encompasses 10,963 meticulously crafted diverse referring expressions for 824 objects spanning 187 categories. Each referred object features an average of 8.3 distinct positive expressions alongside 5.0 negative expressions for non-existent objects. To evaluate model robustness to expression diversity, we propose two complementary metrics: (1) Hard Pass Rate, which necessitates successful localization across all expressions referring to the same object; and (2) Mean Consistency Rate, which quantifies how VLMs generate consistent outputs for expressions describing the same object. Our evaluation reveals that state-of-the-art models struggle with consistent object comprehension. In our assessment, the leading model, Qwen2.5-VL-72B, attains merely 27.7% on Hard Pass Rate and identifies all negative expressions for only 10.1% of images. DRef can serve as a rigorous evaluation suite for assessing the robustness of REC models under diverse expressions, and hopefully encourage efforts toward increasing the reliability of REC systems in real-world applications such as robots. Code and dataset are available in supplementary materials [1].

## 1 Introduction

Referring expression comprehension (REC) challenges models to identify objects in images based on natural language descriptions Kazemzadeh et al. (2014). This task requires sophisticated integration of visual perception and linguistic understanding. Recent vision-language models (VLMs) have demonstrated remarkable progress, with state-of-the-art systems achieving accuracy exceeding 90% on established benchmarks Kazemzadeh et al. (2014); Mao et al. (2015). Notable examples include Grounding DINO Liu et al. (2023e), DeepSeek-VL2 Wu et al. (2024), InternVL series Chen et al. (2024d;c), and Qwen series (Bai et al., 2023; Wang et al., 2024).

Despite impressive benchmark results, we argue that current evaluation protocols are insufficient for assessing models' comprehensive understanding of object recognition due to their limited diversity. They fail to capture the diversity of language used to describe objects in real-world scenarios. Our evaluations reveal a substantial performance gap between the lower and upper bounds within this limited evaluation framework. In the real-world scenarios, a model may encounter diverse and complex referring expressions that vary in linguistic structures, level of details, and description perspectives. *For example, consider the robotic instruction following task* Brohan et al. (2022; 2023);

---

[1] https://anonymous.4open.science/r/DRef-Anonymous-ICLR-AD29/
Due to the size limitation of the supplementary materials, we provide the anonymous link instead

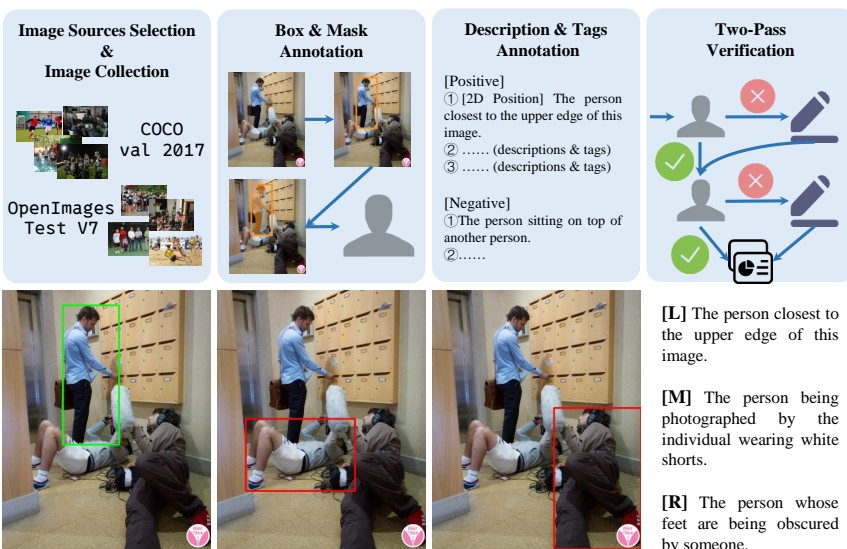

Figure 1: **Upper:** the annotation pipeline of our benchmark. **Lower:** prediction results of Qwen2.5-VL-72B on one object of our benchmark. Referring expressions of the left, middle, and right images are starting with the [L], [M], and [R], respectively.

Chen et al. (2024a), A robot might easily recognize a refrigerator when it's explicitly instructed to do so. However, with the instruction "bring me a can of soda" in a kitchen setting, if the soda is not explicitly placed, the robot should reason about the possible locations of the soda, *e.g.*, the refrigerator. The problem is then equivalent to locate "the place where soda drinks are most likely to appear".

Figure 1 (lower part) demonstrates the inconsistent behavior of the Qwen2.5-VL-72B Bai et al. (2025) when presented with three different referring expressions for the same person in a real image. Despite all expressions referring to the identical individual, the model's performance varies dramatically depending on the expression. While it correctly identifies the target person when given a straightforward spatial description, it fails when the referring expression requires understanding of social interaction context or occlusion relationships. For instance, when prompted to identify "the person being photographed by the individual wearing white shorts", the model incorrectly selects the photographer ("the individual wearing white shorts") rather than the subject of the photograph. This inconsistency across semantically equivalent references to the same object reveals a critical limitation in current models: their comprehension remains highly dependent on the specific expression used, rather than demonstrating robust understanding of the visual scene. This phenomenon indicates that current benchmarks overestimate the performance of models because they lack diversity of expressions.

To address this issue, we introduce **D**iverse **Ref**erring Expression Comprehension (DRef), a novel benchmark designed to evaluate comprehensive understanding of objects in visual scenes. Different from existing benchmarks, DRef provides multiple diverse referring expressions for each target object, intentionally capturing different descriptive perspectives. These expressions span a comprehensive range of referential strategies: object attributes, spatial positioning (both 2D and 3D), relative locations, state descriptions, and expressions requiring reasoning from visual details. By evaluating model performance across this spectrum of referring expressions for identical objects, DRef provides a more rigorous and realistic assessment framework that better reflects the linguistic variation models would encounter in real-world applications.

Hard Pass Rate measures a model's comprehensive understanding by requiring successful localization of a target object across all its diverse referring expressions at a given confidence threshold. While this metric effectively captures a model's ability to understand objects from multiple perspectives, it does not fully characterize the pattern of successes and failures across expressions. For instance, a model that succeeds on all but one expression demonstrates greater reliability than a model that succeeds on only one expression, though both would fail the Hard Pass Rate criterion. To

address this nuance, we introduce Mean Consistency Rate, which evaluates the stability of a model's performance across different expressions for the same object. Together, these metrics provide a more nuanced assessment of visual language models, offering insights into both their comprehensive understanding capabilities and their behavioral consistency.

Our evaluation results demonstrate that DRef presents a substantially more challenging evaluation environment compared to existing REC benchmarks. By requiring models to process diverse perspectives of the same visual entity, it reveals performance gaps that conventional benchmarks fail to detect. Analysis shows that even state-of-the-art models exhibit significant performance degradation when evaluated on their ability to comprehend multiple referring expressions for identical objects, highlighting a critical weakness in current visual language understanding capabilities. This approach bridges an important gap in current evaluation methodologies by more accurately reflecting the heterogeneity of human reference in natural communication contexts. Furthermore, DRef's multi-perspective design enables more nuanced diagnosis of model weaknesses across different referential strategies. To our knowledge, DRef represents the first systematic benchmark specifically designed to evaluate comprehensive object understanding across diverse referring expressions, providing a more reliable indicator of model readiness for deployment in real-world applications where linguistic variation is inevitable.

In summary, our contributions are as follows:

- We propose DRef, a benchmark that evaluates whether vision-language models can handle multiple, diverse referring expressions for the same object, thereby uncovering limitations overlooked by current REC benchmarks.
- We propose two complementary metrics, Hard Pass Rate and Mean Consistency Rate, that quantify robustness to linguistic variation in REC task and evaluate consistency across different perspectives, yielding a more nuanced and realistic assessment of model capabilities.
- Our systematic analysis uncovers a significant performance gap between existing benchmarks and our multi-expression evaluation framework. Notably, state-of-the-art vision-language models that excel on current benchmarks still struggle with comprehensive object understanding under natural variation in human referring expressions.

## 2 RELATED WORK

**Referring Expression Comprehension (REC).** RefCOCO, RefCOCO+ Kazemzadeh et al. (2014), and RefCOCOg Mao et al. (2015) are widely adopted built upon the MS COCO 2014 (Lin et al., 2014). RefCOCO contains approximately 50k annotations across about 20k images, characterized by short and simple expressions. RefCOCO+ increases semantic complexity by excluding locational descriptions. RefCOCOg provides more complex annotations with longer expressions. Described Object Detection Xie et al. (2023) simultaneously considers both the open vocabulary detection Zang et al. (2022); Zareian et al. (2020); Yao et al. (2024); Minderer et al. (2022) and referring expression comprehension, enabling flexible expressions to refer to arbitrary numbers of instances per image. HC-RefLoCo Wei et al. (2024) is specifically designed for grounding with human-centric expressions, which includes about 13,452 images, 24,129 instances, and 44,738 annotations. Human-centric datasets such as HC-RefLoCo Wei et al. (2024) (with 13,452 images, 24,129 instances, and 44,738 annotations) and HumanRef Jiang et al. (2025) focus on grounding person-specific expressions, with the latter emphasizing one-to-many referring relationships (averaging 2.2 instances per referring statement across 103,028 statements). MMR Jang et al. (2025) extends the REC task to locate multiple objects from a single expression. Cops-Ref Chen et al. (2020) incorporates extra images to introduce distracting factors. FineCops-Ref Liu et al. (2024b) further facilitates multi-level reasoning, where each object is categorized into one of three difficulty levels. SCALAR-VG Yang et al. (2024) emphasizes multi-task learning by utilizing bounding boxes, keypoints, and polygons for object localization. Despite these advancements in complexity and reasoning requirements, none specifically evaluate model robustness, a limitation our DRef directly addresses. Our DRef adopts a many-to-one paradigm, where each object is described by many referring expressions from varying perspectives. See Appendix for more details.

**Visual Grounding Models.** Recent advancements in VLMs, *e.g.*, BLIP series Li et al. (2022; 2023), Flamingo Alayrac et al. (2022), InstructBLIP Dai et al. (2023), have significantly im-

Table 1: Comparison between existing benchmarks and our proposed DRef. EPO is the short of "Expressions Per Object". We compare most widely used REC benchmarks, including RefCOCO, RefCOCO+ Kazemzadeh et al. (2014), RefCOCOg Mao et al. (2015), and ReasonSeg Lai et al. (2023) as they are the widely used REC benchmarks.

| Benchmark | Hard Constraint | High Diversity | # EPO |
|---|---|---|---|
| RefCOCO | ✗ | ✗ | 2.84 |
| RefCOCO+ | ✗ | ✗ | 2.82 |
| RefCOCO+ | ✗ | ✗ | 1.91 |
| ReasonSeg | ✗ | ✗ | 1 |
| DRef (Ours) | ✓ | ✓ | 13.30 |

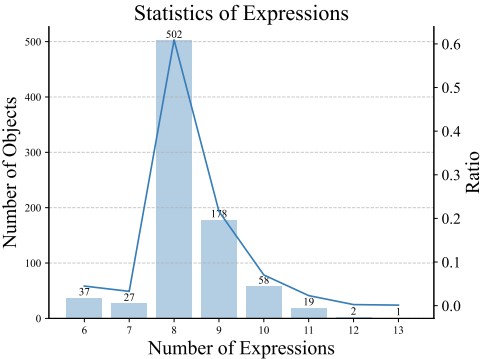

Figure 2: The statistics of expressions of each referred object (positive expressions).

proved the vision-language understanding by leveraging the progress of large language models (LLMs) (Zhang et al., 2022; Chung et al., 2022). The visual grounding task further requires the model to locate the referred object. Some works employ a special head to regress the bounding box of the referent, *e.g.*, ScanFormer Su et al. (2024), M-DGT Chen & Li (2022), and QRNet (Ye et al., 2022). More recently, LLaVA-inspired architectures Liu et al. (2023c;b) have gained prominence, wherein models like Shikra Chen et al. (2023), Ferret You et al. (2024), KOSMOS-2 Peng et al. (2024), Grounding-GPT Li et al. (2024), Cambrian-1 Tong et al. (2024), and SPHINX Lin et al. (2023) connect visual encoders to auto-regressive LLMs and represent bounding boxes as text strings. MRES Wang et al. (2023), OneRef Xiao et al. (2024), LISA Lai et al. (2023); Yang et al. (2023), PixelLLM Ren et al. (2023), PSALM Zhang et al. (2025), GlaMM Rasheed et al. (2024), and GROUNDINHOG Zhang et al. (2024) further explore the more fine-grained localization using the segmentation masks.

## 3 DATASET AND METRICS

Compared to existing benchmarks such as RefCOCO Kazemzadeh et al. (2014) and ReasonSeg Lai et al. (2023), our DRef exhibits greater diversity and enables more rigorous assessment of model robustness through metrics with hard constraints (Table 1). In this section, we present the dataset statistics (Section 3.1) and curation process (Section 3.2) of our DRef benchmark. We then introduce the evaluation metrics designed to assess model robustness and consistency (Section 3.3), where hard constraints are incorporated into the robustness evaluation.

### 3.1 DATASET STATISTICS

**Categories.** Our DRef comprises 824 objects distributed across 187 categories. We group them into 12 general concepts and visualize the hierarchy in the appendix (Figure 15). More details and statistics are available in the appendix (Section A.8).

**Referring Expressions.** DRef consists of 10,963 challenging referring expressions, comprising 6,855 positive expressions (describing existent objects) and 4,108 negative expressions (describing non-existent objects). Figure 2 illustrates that the majority of referred objects are annotated with at least eight diverse expressions. For each image, there are an average of five negative expressions describing non-existent objects. Examples are presented in the appendix (Section A.7).

**Tags & Diversity.** Each referring expression is tagged with at least one describing perspective. Based on these tags, we can annotate diverse referring expressions for each object. We define 9 tags for positive referring expressions, and 1 negative tag for expressions of non-existent objects. Table 2 presents the definitions and examples of these tags. As an example, "The person who is speaking to the woman in red" contains both the "interaction" tag and the "attribute" tag. Figures 3 and 4 visualize the distribution of positive expressions and objects across each tag, respectively. Based on Figure 4, we can see that most objects have at least 7 tags of positive expressions, indicating the high diversity of expressions in our benchmark.

Table 2: The definition as well as the example of the tags in DRef.

| Tag | Description | Example |
|---|---|---|
| 2D Position | The position description on a plane, *e.g.*, image plane. | The person closest to the top of the image. |
| 3D Position | The position description on a 3D space. | The bird in the mid-air. |
| Relative Position | Describe the position of the object relative to other objects. | The person to the left of the car. |
| Size | The size of the object, typically compared with other objects. | The visually largest toy. |
| Attribute | The intrinsic properties of the object. | The red car. |
| Interaction | The interaction between the object and other objects or the environment. | The person who is speaking to the woman in red. |
| Possible Usage | The possible usage or designed purpose of the object. | The object that is used for holding garbage. |
| Possible Status | The possible status of the object. | The device that is turned on. |
| Reasoning | Requires reasoning from existing visual and textual cues. | The person most likely to be the band's lead singer. |
| Negative | The object that is not in the image. | The person who is speaking to the woman in green. |

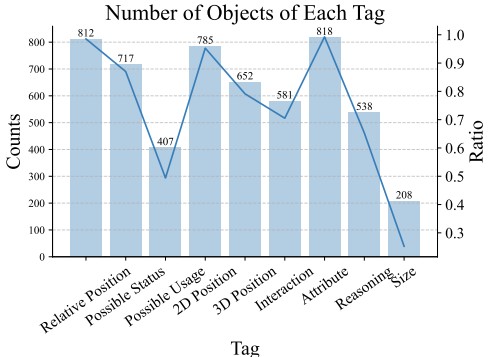

Figure 3: The statistics of positive referring expressions of each tag.

Figure 4: The statistics of referred objects of each tag.

## 3.2 DATASET CURATION

As illustrated in Figure 1, we design an annotation pipeline to decouple the highly difficult and time-consuming process into several steps: (1) image source selection, (2) image collection, (3) box and mask annotation, (4) object description annotation, and (5) two-pass verification. Steps (4) and (5) proved most challenging and labor-intensive, requiring annotators to formulate unambiguous and uniquely identifying descriptions for each object. On average, annotating all positive and negative expressions for a single image required over 30 minutes. Utilizing this structured pipeline, our team of seven annotators produced 10,963 expressions across 824 images over two months.

**Image Sources Selection and Image Collection.** We first determine which data sources the images should be extracted from. We first randomly sample a few images of validation and test sets from existing image datasets, *e.g.*, Flicker30K Young et al. (2014), MS COCO Lin et al. (2014), and OpenImages Kuznetsova et al. (2020), etc. Then, annotators evaluate these samples for scene complexity, and we rank data sources based on the proportion of complex images they contain. The images for further annotation are extracted from the most complex data source until it runs out. Through this process, our images are collected from the MS COCO val 2017 and OpenImages V7 test datasets. We excluded images with simple scenes, such as close-ups of single objects or scenes containing only one object category. Detailed selection criteria are provided in the appendix (Section A.10).

**Box and Mask Annotation.** For each selected object, annotators label its bounding box. Using these boxes as input, we employ the SAM-2.1-H model Ravi et al. (2025) to automatically generate segmentation masks. Each mask underwent quality assurance review by annotators, who manually correct issues such as internal holes or the inclusion of non-object areas. When the model-generated masks are substantially inaccurate, annotators create new masks manually to ensure precise object delineation. We ask annotators to label only one object per image to improve scene diversity.

**Object Description Annotation.** During this critical phase, annotators generate a minimum of six diverse referring expressions for each selected object. To ensure description diversity, we instruct

annotators to characterize objects from different perspectives that are listed in Table 2. Perspectives inapplicable to specific objects could be omitted. Throughout this process, annotators verify that each referring expression uniquely and unambiguously identifies its target object. As for the negative expressions that refer to non-existent objects, the core idea is to try to fool the model. Annotators are instructed to: (1) change the modifiers in a sentence, *e.g.*, shape, color, size, positions, etc; (2) increase or decrease the number of modifiers; (3) modify other fine-grained details. For more details, please refer to the appendix (Section A.10).

**Quality Control.** We implement a two-pass quality control process to ensure annotation accuracy and clarity. During each pass, annotators verify that all referring expressions are unique and unambiguous. When they detect ambiguity in a description, they modify it to eliminate any potential confusion. Similarly, if they found that a referring expression could apply to multiple objects in the image, they refine the description to ensure it uniquely identifies the intended target object. This iterative verification process was crucial for maintaining the benchmark's integrity and utility for evaluating referring expression comprehension models.

## 3.3 EVALUATION METRICS

We first introduce the concept of successful localization: (1) located expression: for a given expression, we said the expression is successfully located if the Intersection over Union (IoU) between the predicted bounding box and the ground truth bounding box exceeds the threshold; (2) located object: for a given object, we said the object is successfully located if all the expressions referring to the object are successfully located; (3) partially located object: for a given object, at least one of the expressions referring to the object is successfully located. For the negative expressions of an image, we define a special category of "non-existent object". In these cases, we assign an IoU of 1 if the model correctly identifies the object as "not in the image", and 0 otherwise.

Besides the mean precision $P$ that defined as the proportion of the number of expressions successfully located, we also propose two metrics, Hard Pass Rate and Mean Consistency Rate, to evaluate a model's performance in locating referred objects across diverse expressions. The proposed metrics specifically assess robustness and output consistency at the object level while accounting for linguistic variations.

**Metrics: Hard Pass Rate.** We said the model passes the test of the object if the object is successfully located. The Hard Pass Rate metric calculates the proportion of objects that are successfully located, which requires successful localization across all diverse expressions referring to the same object, or correct recognition of all negative expressions for non-existent objects.

We also consider a soft version of Hard Pass Rate, namely Soft Pass Rate, to highlight the drawback of the limited diversity and the absence of hard constraints of metric in existing benchmarks. The Soft Pass Rate calculates the proportion of objects that are partially located. By randomly extracting individual expressions from our benchmark, we can construct multiple "virtual benchmarks" that simulate these existing benchmarks. The performance of a model on these virtual benchmarks could vary significantly depending on the selected expression. The Hard Pass Rate and Soft Pass Rate thus serve as a lower bound and upper bound of such virtual benchmarks, respectively. We demonstrate in Section 4 that the gap between the two bounds is significant, indicating that the existing benchmarks are problematic and a more comprehensive evaluation is necessary.

Denoting the set of all diverse expressions of an object as $\mathcal{E}$, the set of all objects as $\mathcal{O}$, the proposed metrics are defined as follows:

$$R^{\text{Hard}} = \frac{1}{|\mathcal{O}|} \sum_{o}^{|\mathcal{O}|} \mathbf{1}[(\sum_{e}^{|\mathcal{E}|} s_{o,e}) \geq |\mathcal{E}|] \qquad (1)$$

$$R^{\text{Soft}} = \frac{1}{|\mathcal{O}|} \sum_{o}^{|\mathcal{O}|} \mathbf{1}[(\sum_{e}^{|\mathcal{E}|} s_{o,e}) \geq 1] \qquad (2)$$

where the threshold $\tau$ is omitted for simplicity. $\mathbf{1}[\cdot]$ is the indicator function. $s_{o,e}$ is interpreted as success and is defined as 1 if either a model: (1) successfully locates object $o$ referred by expression $e$, or (2) successfully recognizes that the referred object does not exist; otherwise, $s_{o,e}$ equals 0.

In our evaluation, we mainly set the $\tau$ as 50% ($R_{50}^{\text{Hard}}$ and $R_{50}^{\text{Soft}}$). The value of $\tau$ can be increased to meet the requirements for higher localization accuracy, *e.g.*, $R_{75}^{\text{Hard}}$ and $R_{95}^{\text{Hard}}$.

Importantly, high Soft Pass Rate does not imply high Hard Pass Rate. Consider an extreme case where, for each object, the model consistently predicts exactly one incorrect result for each expression. This would produce a Soft Pass Rate of 1 while resulting in a Hard Pass Rate of 0.

**Metrics: Mean Consistency Rate.** While the $R^{\text{Hard}}$ metric evaluates overall localization accuracy, it cannot assess whether a model consistently locates objects across different referring expressions. To further evaluate the consistency, we propose the mean consistency rate $R^{\text{C}}$, which measures prediction stability across diverse expressions for each object in the benchmark. For each object, the similarity $S_{i,j}$ of each pair of predicted bounding boxes is calculated by their IoU. We then build a similarity matrix $S$ of size $|\mathcal{E}| \times |\mathcal{E}|$ for all predictions. The consistency rate $\hat{R}_o^{\text{C}}$ for object $o$ is calculated as the mean of the strictly upper (or lower) triangular elements in matrix $S$:

$$\hat{R}_o^{\text{C}} = \frac{2}{|\mathcal{E}|(|\mathcal{E}| - 1)} \sum_{i=1}^{|\mathcal{E}|} \sum_{j=i+1}^{|\mathcal{E}|} S_{i,j} \tag{3}$$

However, the naive consistency rate can be misleading if a model consistently produces incorrect predictions, *e.g.*, consistently localizing a human when it's asked to locate a car. To address this, we introduce a modified consistency rate that incorporates the mean IoU between the prediction and the ground truth. Specifically, denoting the IoU between the prediction and the ground truth as $\text{IoU}_{o,e}$, we weight the consistency rate of each object by the average of the IoUs of that object, and calculate the mean across all objects:

$$R_o^{\text{C}} = \frac{\sum_e^{|\mathcal{E}|} \text{IoU}_{o,e}}{|\mathcal{E}|} \cdot \hat{R}_o^{\text{C}}, \quad R^{\text{C}} = \frac{1}{|\mathcal{O}|} \sum_o^{|\mathcal{O}|} R_o^{\text{C}} \tag{4}$$

If $R^{\text{C}}$ is close to 1, it indicates that the model consistently produces similar bounding boxes for different expressions referring to the same object, demonstrating strong consistency in object comprehension. Conversely, a low $R^{\text{C}}$ suggests that the model's predictions vary significantly across different expressions, or that the model struggles to accurately locate the referred object, indicating poor consistency in object comprehension.

## 4 BENCHMARKING RESULTS AND ANALYSIS

### 4.1 MODELS AND METRICS

We evaluate a range of state-of-the-art vision-language models: PolyFormer Liu et al. (2023d), MDETR Kamath et al. (2021), ReLA Liu et al. (2023a), X-Decoder Zou et al. (2022), SEEM Zou et al. (2023), LISA Lai et al. (2023), InternVL series Chen et al. (2024b); Zhu et al. (2025), and Qwen2.5-VL series Bai et al. (2025). Full results of all evaluated models are provided in the appendix (Table 7). Additionally, we also include recent findings VLM-R1-3B Shen et al. (2025), which enhances Qwen2.5-VL-3B through reinforcement fine-tuning (RFT) (DeepSeek-AI et al., 2025; Shao et al., 2024). We generate corresponding masks using SAM-2.1-H Ravi et al. (2025) for models that produce only bounding boxes, *e.g.*, Qwen2.5-VL series Bai et al. (2025). Conversely, for models that output only masks, we derive bounding boxes from their mask outputs. For the expected output format of negative expressions, please check the appendix (Section A.2.1).

We also perform two-stage fine-tuning on Qwen2.5-VL-7B using 368 *additionally labeled objects* as training data, following the same pipeline as DRef. Specifically, we apply supervised fine-tuning followed by reinforcement fine-tuning via GRPO (DeepSeek-AI et al., 2025; Shao et al., 2024) on the same training data. We denote this model as Qwen2.5-VL-7B-SFT-RFT.

Table 3: **The performance of the advanced vision-language models on DRef.** The SEEM models with subscript $_S$ means it only support single interactive inference, while the models with $_M$ support multiple interactive inference.

| Model | Bbox | | | | | | Mask | | | | | |
| | Positive | | | | Negative | | Positive | | | | Negative | |
| | $R_{50}^{\text{Hard}}$ | $P_{50}$ | $R_{50}^{\text{Soft}}$ | $R^{\text{C}}$ | $R^{\text{Hard}}$ | $P$ | $R_{50}^{\text{Hard}}$ | $P_{50}$ | $R_{50}^{\text{Soft}}$ | $R^{\text{C}}$ | $R^{\text{Hard}}$ | $P$ |
|---|---|---|---|---|---|---|---|---|---|---|---|---|
| OpenAI GPT-5 | 0.7 | 9.2 | 30.0 | 9.0 | 1.1 | 24.8 | 0.7 | 14.5 | 45.1 | 6.1 | 1.1 | 25.3 |
| OpenAI o4-mini | 2.5 | 27.1 | 56.8 | 13.3 | 6.8 | 53.0 | 3.2 | 35.6 | 77.8 | 13.1 | 7.0 | 53.2 |
| Gemini-1.5-Flash | 2.9 | 5.7 | 8.6 | 8.9 | 0.8 | 12.4 | 2.9 | 6.1 | 11.3 | 4.3 | 1.0 | 12.6 |
| Gemini-2.5-Pro | 3.2 | 6.6 | 9.6 | 12.3 | 1.3 | 26.7 | 3.6 | 8.1 | 12.7 | 6.5 | 1.3 | 26.9 |
| Gemini-3-Pro | 3.3 | 6.7 | 7.8 | 12.8 | 0.4 | 21.3 | 5.0 | 7.9 | 9.1 | 6.9 | 0.4 | 21.4 |
| PolyFormer-L | 8.3 | 51.2 | 78.4 | 28.6 | 0.0 | 0.0 | 8.0 | 49.2 | 75.6 | 24.2 | 0.0 | 0.0 |
| MDETR | 7.5 | 49.4 | 84.3 | 25.9 | 0.7 | 11.3 | 7.3 | 49.3 | 85.6 | 26.8 | 0.7 | 11.3 |
| ReLA | 2.3 | 34.1 | 67.6 | 15.8 | 5.0 | 18.0 | 2.8 | 32.8 | 66.4 | 13.4 | 5.0 | 18.0 |
| X-Decoder-L | 5.5 | 37.0 | 67.6 | 20.1 | 1.6 | 11.0 | 5.1 | 36.7 | 68.9 | 18.5 | 1.6 | 11.0 |
| SEEM-L$_S$ | 5.7 | 38.8 | 69.1 | 21.2 | 1.2 | 8.2 | 5.1 | 37.9 | 68.9 | 19.6 | 1.2 | 8.2 |
| SEEM-L$_M$ | 6.7 | 39.4 | 68.6 | 22.4 | 1.3 | 6.4 | 6.1 | 38.8 | 69.3 | 20.9 | 1.3 | 6.4 |
| SEEM-SAM-L | 8.5 | 35.0 | 66.0 | 21.7 | 0.6 | 0.6 | 8.4 | 34.0 | 64.4 | 19.4 | 0.6 | 0.6 |
| LISA-7B | 3.8 | 34.2 | 68.0 | 20.5 | 0.0 | 1.3 | 3.9 | 41.5 | 78.9 | 18.9 | 0.0 | 1.3 |
| LISA-13B | 6.7 | 42.9 | 75.5 | 25.0 | 0.0 | 1.0 | 7.6 | 48.5 | 82.8 | 24.8 | 0.0 | 1.0 |
| GLaMM | 6.7 | 47.9 | 74.8 | 28.6 | 0.0 | 0.3 | 9.0 | 50.1 | 78.5 | 27.7 | 0.0 | 0.3 |
| Llava-Next-Mistral-7B | 15.8 | 53.6 | 74.3 | 31.0 | 0.0 | 0.8 | 16.9 | 57.5 | 84.0 | 35.9 | 0.0 | 0.8 |
| Llava-Next-Llama3-8B | 12.4 | 49.7 | 74.9 | 27.9 | 0.0 | 0.5 | 13.0 | 51.6 | 80.8 | 31.0 | 0.0 | 0.5 |
| InternVL2.5-8B | 12.3 | 48.6 | 71.8 | 23.8 | 1.6 | 20.8 | 12.1 | 51.2 | 78.9 | 30.1 | 1.6 | 20.8 |
| InternVL3-8B | 22.1 | 66.4 | 89.9 | 42.2 | 0.1 | 3.5 | 21.5 | 66.5 | 91.1 | 45.1 | 0.1 | 3.5 |
| Qwen2.5-VL-7B | 18.4 | 57.8 | 88.0 | 35.7 | 0.0 | 0.0 | 17.6 | 56.7 | 87.3 | 36.6 | 0.0 | 0.0 |
| Qwen2.5-VL-32B | 23.7 | 63.6 | 96.1 | 46.4 | 0.2 | 10.2 | 22.5 | 70.6 | 95.5 | 47.8 | 0.2 | 10.2 |
| Qwen2.5-VL-72B | 27.7 | 77.4 | 97.2 | 51.4 | 10.1 | 57.6 | 26.2 | 76.5 | 96.0 | 54.4 | 10.1 | 57.6 |
| Qwen3-VL-32B | 32.8 | 78.1 | 97.5 | 53.8 | 0.5 | 21.1 | 32.5 | 77.6 | 96.6 | 57.2 | 0.5 | 21.1 |
| VLM-R1-3B | 27.2 | 38.9 | 51.2 | 33.8 | 0.0 | 0.0 | 26.6 | 37.9 | 50.7 | 31.9 | 0.0 | 0.0 |
| Qwen2.5-VL-7B-SFT-RFT | 30.0 | 74.7 | 94.1 | 52.2 | 0.0 | 0.0 | 29.5 | 74.3 | 93.4 | 68.5 | 0.0 | 0.0 |

As mentioned in Section 3.3, we employ multiple metrics to evaluate the performance of models on our DRef, including Hard Pass Rate, Soft Pass Rate, Mean Consistency Rate, and mean precision $P$. For pass rates and precision, we set the IoU threshold to 0.5 and calculate $R_{50}^{\text{Hard}}$, $P_{50}$, and $R_{50}^{\text{Soft}}$ accordingly.

## 4.2 MAIN RESULTS

As illustrated in Table 3, our benchmark DRef reveals that even advanced VLMs such as X-Decoder and SEEM struggle to consistently identify referred objects across diverse expressions. Although most models report relatively good results on $P_{50}$, the performance on $R_{50}^{\text{Hard}}$ is significantly worse. Notably, Qwen2.5-VL-72B, the most powerful model in our evaluation, achieves only 27.7% on $R_{50}^{\text{Hard}}$, suggesting poor robustness confronted with diverse expressions.

**Analysis on Qwen-VL Series.** Our evaluation across different model scales in the Qwen2.5-VL series reveals clear performance trends on multiple metrics. With the model size increasing, there are substantial gains on the $P_{50}$ and $R_{50}^{\text{Soft}}$. The 7B model in Qwen2.5-VL series achieves 18.4% on $R_{50}^{\text{Hard}}$, 57.8% on $P_{50}$, and 88.0% on $R_{50}^{\text{Soft}}$. As model size increases to 32B and 72B, the scores on $R_{50}^{\text{Hard}}$ are improved to 23.7% and 27.7%, respectively, However, the considerably lower Mean Consistency Rate, *e.g.*35.7% of the 7B model and the 46.4% of the 72B model, suggests the model might suffer from hallucination problems and be distracted from other instances in the expressions. We also observe that the ability to reject all negative referring expressions in an image emerges with the 72B model, achieving 10.1% on the $R^{\text{Hard}}$ for negative referring expressions.

**Discussion on Frontier Multi-modal Models.** We observe that GPT-5 [2], o4-mini [3], Gemini-2.5-Pro Team (2025) and Gemini-3-Pro yield poor results. For instance, on the $R_{50}^{\text{Hard}}$ benchmark for the grounding task, these models attain only 0.7%, 2.5%, 3.2%, and 3.3%, respectively, which is

---

[2]https://openai.com/index/introducing-gpt-5/

[3]https://openai.com/index/introducing-o3-and-o4-mini/

Table 5: **Comparison with Existing Expression Benchmarks.** We compare the performance of state-of-the-art models on our DRef benchmark and existing benchmarks: Ref-COCO(val/testA/testB), RefCOCO+(val/testA/testB), and RefCOCOg(val/test). We also present the mIoU of all positive referring expressions following the existing benchmarks. The results show that DRef is more challenging and models are struggling to keep robust.

| Models | DRef $R_{50}^{\text{hard}}$ | DRef $P_{50}$ | DRef mIoU | RefCOCO | RefCOCO+ | RefCOCOg |
|---|---|---|---|---|---|---|
| **Bbox** | | | | | | |
| PolyFormer-L | 8.3 | 51.2 | 48.3 | 90.4 / 92.9 / 87.2 | 85.0 / 89.8 / 78.0 | 85.8 / 85.9 |
| Qwen2.5-VL-72B | 27.7 | 77.4 | 70.8 | 92.7 / 94.6 / 89.7 | 88.9 / 92.2 / 83.7 | 89.9 / 90.3 |
| InternVL2.5-8B | 12.3 | 48.6 | 43.1 | 90.3 / 94.5 / 85.9 | 85.2 / 91.5 / 78.8 | 86.7 / 87.6 |
| InternVL3-8B | 22.1 | 66.4 | 61.0 | 92.5 / 94.6 / 88.0 | 88.2 / 92.5 / 81.8 | 89.6 / 90.0 |
| **Mask** | | | | | | |
| PolyFormer-L | 8.0 | 49.2 | 42.6 | 76.0 / 78.3 / 73.3 | 69.3 / 74.6 / 61.9 | 69.2 / 70.2 |
| LISA-7B | 3.9 | 41.5 | 40.0 | 74.9 / 79.1 / 72.3 | 65.1 / 70.8 / 58.1 | 67.9 / 70.6 |

significantly lower than the performance of other models (Table 3). We hypothesize that this stems from the fact that these frontier closed-source models may not have been specifically optimized for visual grounding tasks. Additionally, though these models are relatively good at identifying negative referring expressions ($P$), they still struggle to reject all negative expressions in an image, as indicated by their low $R^{\text{Hard}}$ scores of 1.1%, 6.8%, 1.3%, and 0.4%, respectively.

**Reinforcement Fine-tuning Improves Qwen2.5-VL-7B.** Our two-stage fine-tuning on Qwen2.5-VL-7B with 368 *additional labeled objects* yields significant performance improvements. Specifically, our Qwen2.5-VL-7B-SFT-RFT model achieves 30.0 on $R_{50}^{\text{hard}}$, surpassing both the 7B baseline (18.4) and the larger 72B model (27.7). While SFT alone improves $P_{50}$ and $R^C$ at the cost of $R_{50}^{\text{hard}}$, the two-stage process yields the highest gains across all metrics ($R_{50}^{\text{hard}}$ +11.6). We conclude that SFT captures object details while RFT enhances robustness. Overall, these results indicate that reinforcement fine-tuning is an effective strategy to enhance the robustness of VLMs against diverse referring expressions.

$R_{50}^{\text{Hard}}$ and $R_{50}^{\text{Soft}}$ **Reveal the Limitation of Existing benchmarks.** As stated in Section 3.3, $R_{50}^{\text{Hard}}$ and $R_{50}^{\text{Soft}}$ serve as the lower and upper bounds of the performance of a model on the simulated benchmarks missing diverse expressions and hard metric constraints. Taking Qwen2.5-VL-72B as an example (Table 3), we see that there is a huge gap between the $R_{50}^{\text{Hard}}$ and $R_{50}^{\text{Soft}}$ scores, *i.e.*, 27.7% and 97.2%, indicating that the different expressions can lead to significantly different results, hence showing that evaluation on benchmarks without diversity and hard metric constraints are not robust and reliable.

Table 4: **Two-stage fine-tuning results on Qwen2.5-VL-7B.**

| Models | $R_{50}^{\text{Hard}}$ | $P_{50}$ | $R_{50}^{\text{Soft}}$ | $R^C$ |
|---|---|---|---|---|
| baseline | 18.4 | 57.8 | 88.0 | 35.7 |
| + SFT | 16.9 | 66.3 | 94.2 | 39.8 |
| $\Delta$ | -1.5 | +8.5 | +6.2 | +4.1 |
| + RFT | 24.5 | 74.7 | 95.8 | 50.5 |
| $\Delta$ | +6.1 | +16.9 | +7.8 | +14.8 |
| + SFT + RFT | 30.0 | 74.7 | 94.1 | 52.2 |
| $\Delta$ | +11.6 | +16.9 | +6.1 | +16.5 |

**Comparison with Existing Expression Benchmarks.** As shown in Table 5, the experimental results demonstrate that our DRef benchmark is significantly more challenging than existing expression benchmarks due to its diverse referring expressions. Current state-of-the-art models, which achieve high performance on traditional benchmarks, show substantial performance drops on DRef, *i.e.*, $R_{50}^{\text{Hard}}$ and $P_{50}$, highlighting the concern of the stability of these models when facing diverse referring expressions in real-world scenarios.

**Graded Hard Pass Rate Analysis.** We further analyze the model performance using the graded hard pass rate, as shown in Figure 5. For a given percentage $p$, an object is considered successfully localized if the model correctly identifies it in at least $p\%$ of the referring expressions associated with that object. This metric provides a nuanced view of model performance across varying levels of expression difficulty. The gentle decrease in performance indicates that models are relatively robust for approximately 60% of the expressions in an object, suggesting their difficulty is consistent. As the threshold further increases, the models' performance suddenly drops, indicating that the remaining expressions are difficult for the model.

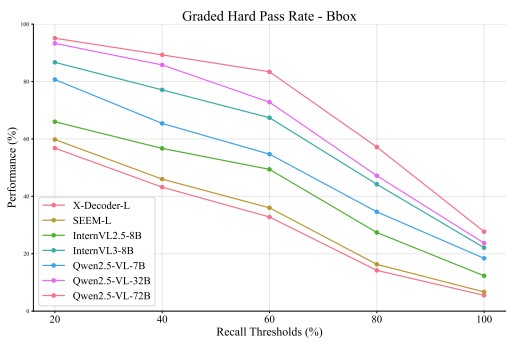
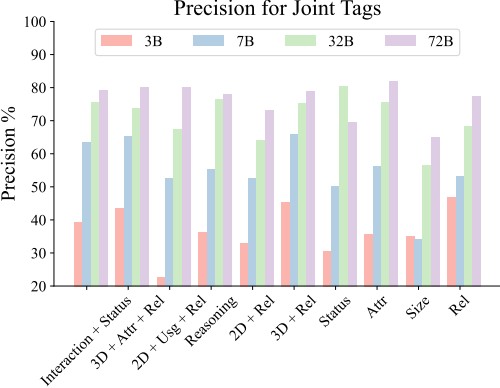

Figure 5: **Graded Hard Pass Rate.** We analyze the model performance using the graded hard pass rate. When facing more challenging referring expressions, the performance of models drop more significantly.

Figure 6: **The precision of Qwen2.5-VL series of the different composition of tags in DRef.** Status: Possible Status; 2D: 2D Position; 3D: 3D Position; Attr: Attribute; Rel: Relative Position; Usg: Possible Usage;

**Precision on Different Tag Groups.** We visualize the precision across different tag groups in Figure 6. We prioritized the 10 most frequent tag combinations in the dataset. Qwen2.5-VL-72B demonstrates remarkably balanced precision across all examined tag categories. In contrast, Qwen2.5-VL-3B exhibits substantial performance degradation, specifically on expressions that simultaneously incorporate 2D Position, Possible Usage, and Relative Position tags. This reveals a weakness of Qwen2.5-VL-3B in handling expressions with special tag combinations.

**Influence of Linguistic Styles.** Our labeled referring expressions primarily focus on the visual dimension, where a single object within a current visual scene can be referred to using various descriptive perspectives. To investigate the impact of linguistic style on model performance, we employed GPT-5 to rewrite all referring expressions into "colloquial" and "reordered" forms while preserving the original semantics. The colloquial style mimics natural speech patterns, whereas the reordered style modifies syntax without altering the core meaning. As shown in Table 6, although rewriting the original descriptive texts posed a certain degree of challenge to the models under evaluation, it did not result in significant performance degradation. Nonetheless, the models still struggle to maintain robustness when confronted with diverse referring expressions, regardless of the linguistic style employed.

Table 6: **Comparing rewritten expressions with different linguistic styles on Qwen2.5-VL-7B.**

| | **Positive** | | | | **Negative** | |
|---|---|---|---|---|---|---|
| **Text Style** | $R_{50}^{\text{Hard}}$ | $P_{50}$ | $R_{50}^{\text{Soft}}$ | $R^{\text{C}}$ | $R^{\text{Hard}}$ | $P$ |
| Original | 18.4 | 57.8 | 88.0 | 35.7 | 0.0 | 0.0 |
| Colloquial | 17.5 | 60.0 | 90.3 | 36.8 | 0.0 | 0.0 |
| Reordered | 17.2 | 59.3 | 90.3 | 36.2 | 0.0 | 0.0 |

## 5 CONCLUSION

We propose a new benchmark DRefas well as two novel metrics, Hard Pass Rate and Mean Consistency Rate, for the referring expression comprehension. DRef encompasses 10,963 carefully annotated referring expressions for 824 objects from the MS COCO val 2017 and OpenImages V7 test datasets. The Hard Pass Rate is designed to evaluate the model's robustness under diverse and complex referring expressions, while the Mean Consistency Rate metric is designed to evaluate the consistency of the model on locating the referred object over diverse expressions. Our evaluation demonstrated that DRef presented challenges for VLMs. These models underperform in both robustness and consistency when challenged with diverse expressions. While our DRef focus on the multiple diverse referring expressions for each object, we suppose that the complementary problem where a single referring expression can refer to multiple objectsis also significant. We leave the exploration of diverse expressions for multiple objects as future work.

## 6 REPRODUCIBILITY STATEMENT

We present the details of curation of our benchmark in Section 3.2, and provide the annotation guidelines in Appendix A.10.

We also provide the dataset, including images and annotations, in the supplementary materials. The code for evaluation is also provided in the supplementary materials.

## 7 USAGE OF LLM

In our experiments, we evaluated some models that incorporate large language models (LLMs), such as Qwen2.5-VL series and InternVL2.5/3 series.

Also, LLMs are used during the writing of this manuscript to improve the readability. We carefully check the generated content to ensure its correctness.

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

Table 7: **The performance of the advanced vision-language models on DRef.** The SEEM models with subscript $_S$ means it only support single interactive inference, while the models with $_M$ support multiple interactive inference.

| Model | Bbox | | | | | | Mask | | | | | |
|---|---|---|---|---|---|---|---|---|---|---|---|---|
| | Positive | | | | Negative | | Positive | | | | Negative | |
| | $R_{50}^{\text{Hard}}$ | $P_{50}$ | $R_{50}^{\text{Soft}}$ | $R^{\text{C}}$ | $R^{\text{Hard}}$ | $P$ | $R_{50}^{\text{Hard}}$ | $P_{50}$ | $R_{50}^{\text{Soft}}$ | $R^{\text{C}}$ | $R^{\text{Hard}}$ | $P$ |
| OpenAI GPT-5 | 0.7 | 9.2 | 30.0 | 9.0 | 1.1 | 24.8 | 0.7 | 14.5 | 45.1 | 6.1 | 1.1 | 25.3 |
| OpenAI o4-mini | 2.5 | 27.1 | 56.8 | 13.3 | 6.8 | 53.0 | 3.2 | 35.6 | 77.8 | 13.1 | 7.0 | 53.2 |
| Gemini-1.5-Flash | 2.9 | 5.7 | 8.6 | 8.9 | 0.8 | 12.4 | 2.9 | 6.1 | 11.3 | 4.3 | 1.0 | 12.6 |
| Gemini-2.5-Pro | 3.2 | 6.6 | 9.6 | 12.3 | 1.3 | 26.7 | 3.6 | 8.1 | 12.7 | 6.5 | 1.3 | 26.9 |
| Gemini-3-Pro | 3.3 | 6.7 | 7.8 | 12.8 | 0.4 | 21.3 | 5.0 | 7.9 | 9.1 | 6.9 | 0.4 | 21.4 |
| PolyFormer-B | 8.1 | 50.6 | 78.6 | 27.6 | 0.0 | 0.0 | 8.0 | 48.7 | 76.0 | 23.6 | 0.0 | 0.0 |
| PolyFormer-L | 8.3 | 51.2 | 78.4 | 28.6 | 0.0 | 0.0 | 8.0 | 49.2 | 75.6 | 24.2 | 0.0 | 0.0 |
| MDETR | 7.5 | 49.4 | 84.3 | 25.9 | 0.7 | 11.3 | 7.3 | 49.3 | 85.6 | 26.8 | 0.7 | 11.3 |
| ReLA | 2.3 | 34.1 | 67.6 | 15.8 | 5.0 | 18.0 | 2.8 | 32.8 | 66.4 | 13.4 | 5.0 | 18.0 |
| X-Decoder-T | 4.1 | 33.5 | 62.4 | 17.9 | 1.6 | 10.7 | 4.1 | 33.8 | 65.0 | 16.4 | 1.6 | 10.7 |
| X-Decoder-L | 5.5 | 37.0 | 67.6 | 20.1 | 1.6 | 11.0 | 5.1 | 36.7 | 68.9 | 18.5 | 1.6 | 11.0 |
| SEEM-T$_S$ | 5.8 | 35.9 | 65.9 | 19.2 | 0.7 | 8.0 | 5.5 | 35.9 | 67.0 | 18.1 | 0.7 | 8.0 |
| SEEM-L$_S$ | 5.7 | 38.8 | 69.1 | 21.2 | 1.2 | 8.2 | 5.1 | 37.9 | 68.9 | 19.6 | 1.2 | 8.2 |
| SEEM-T$_M$ | 4.7 | 34.5 | 65.5 | 18.5 | 1.3 | 7.8 | 4.6 | 34.4 | 66.6 | 17.1 | 1.3 | 7.8 |
| SEEM-L$_M$ | 6.7 | 39.4 | 68.6 | 22.4 | 1.3 | 6.4 | 6.1 | 38.8 | 69.3 | 20.9 | 1.3 | 6.4 |
| SEEM-SAM-B | 9.1 | 35.7 | 64.4 | 22.4 | 0.6 | 0.7 | 8.6 | 34.7 | 63.1 | 20.2 | 0.6 | 0.7 |
| SEEM-SAM-L | 8.5 | 35.0 | 66.0 | 21.7 | 0.6 | 0.6 | 8.4 | 34.0 | 64.4 | 19.4 | 0.6 | 0.6 |
| LISA-7B | 3.8 | 34.2 | 68.0 | 20.5 | 0.0 | 1.3 | 3.9 | 41.5 | 78.9 | 18.9 | 0.0 | 1.3 |
| LISA-13B | 6.7 | 42.9 | 75.5 | 25.0 | 0.0 | 1.0 | 7.6 | 48.5 | 82.8 | 24.8 | 0.0 | 1.0 |
| LISA++-7B | 3.9 | 36.3 | 73.7 | 17.8 | 0.0 | 3.5 | 2.8 | 40.4 | 80.1 | 16.8 | 0.0 | 3.5 |
| GLaMM | 6.7 | 47.9 | 74.8 | 28.6 | 0.0 | 0.3 | 9.0 | 50.1 | 78.5 | 27.7 | 0.0 | 0.3 |
| Llava-Next-Mistral-7B | 15.8 | 53.6 | 74.3 | 31.0 | 0.0 | 0.8 | 16.9 | 57.5 | 84.0 | 35.9 | 0.0 | 0.8 |
| Llava-Next-Llama3-8B | 12.4 | 49.7 | 74.9 | 27.9 | 0.0 | 0.5 | 13.0 | 51.6 | 80.8 | 31.0 | 0.0 | 0.5 |
| DeepSeek-VL2-Small | 25.2 | 60.7 | 84.6 | 43.7 | 0.0 | 0.0 | 24.6 | 60.0 | 84.2 | 43.8 | 0.0 | 0.0 |
| DeepSeek-VL2 | 22.7 | 58.9 | 83.3 | 40.5 | 0.0 | 0.0 | 22.2 | 57.7 | 82.3 | 40.8 | 0.1 | 0.1 |
| InternVL2.5-2B | 2.8 | 23.0 | 49.4 | 11.5 | 1.6 | 0.1 | 2.4 | 23.0 | 54.1 | 10.0 | 0.0 | 0.1 |
| InternVL2.5-4B | 11.8 | 47.5 | 70.4 | 29.6 | 0.1 | 2.3 | 10.6 | 48.0 | 76.6 | 30.0 | 0.1 | 0.0 |
| InternVL2.5-8B | 12.3 | 48.6 | 71.8 | 23.8 | 1.6 | 20.8 | 12.1 | 51.2 | 78.9 | 30.1 | 1.6 | 20.8 |
| InternVL3-1B | 14.0 | 37.2 | 67.6 | 24.8 | 0.1 | 0.9 | 13.1 | 37.3 | 70.9 | 24.4 | 0.0 | 0.0 |
| InternVL3-2B | 14.1 | 54.9 | 80.3 | 33.2 | 0.0 | 1.2 | 13.2 | 54.4 | 81.1 | 33.6 | 0.0 | 0.0 |
| InternVL3-8B | 22.1 | 66.4 | 89.9 | 42.2 | 0.1 | 3.5 | 21.5 | 66.5 | 91.1 | 45.1 | 0.1 | 3.5 |
| Qwen2.5-VL-3B | 26.0 | 37.0 | 48.7 | 32.5 | 0.0 | 0.0 | 24.9 | 35.8 | 47.6 | 30.4 | 0.0 | 0.0 |
| Qwen2.5-VL-7B | 18.4 | 57.8 | 88.0 | 35.7 | 0.0 | 0.0 | 17.6 | 56.7 | 87.3 | 36.6 | 0.0 | 0.0 |
| Qwen2.5-VL-32B | 23.7 | 63.6 | 96.1 | 46.4 | 0.2 | 10.2 | 22.5 | 70.6 | 95.5 | 47.8 | 0.2 | 10.2 |
| Qwen2.5-VL-72B | 27.7 | 77.4 | 97.2 | 51.4 | 10.1 | 57.6 | 26.2 | 76.5 | 96.0 | 54.4 | 10.1 | 57.6 |
| VLM-R1-3B | 27.2 | 38.9 | 51.2 | 33.8 | 0.0 | 0.0 | 26.6 | 37.9 | 50.7 | 31.9 | 0.0 | 0.0 |
| Qwen3-VL-2B | 17.2 | 51.8 | 80.3 | 32.5 | 0.0 | 0.0 | 15.9 | 51.0 | 80.5 | 33.8 | 0.0 | 0.0 |
| Qwen3-VL-4B | 19.5 | 67.8 | 94.2 | 41.9 | 0.0 | 3.2 | 19.7 | 67.7 | 94.2 | 44.5 | 0.0 | 3.2 |
| Qwen3-VL-8B | 18.8 | 67.5 | 95.0 | 40.9 | 0.0 | 1.1 | 19.2 | 67.9 | 95.5 | 44.4 | 0.0 | 1.1 |
| Qwen3-VL-32B | 32.8 | 78.1 | 97.5 | 53.8 | 0.5 | 21.1 | 32.5 | 77.6 | 96.6 | 57.2 | 0.5 | 21.1 |
| Qwen2.5-VL-7B-SFT-RFT | 30.0 | 74.7 | 94.1 | 52.2 | 0.0 | 0.0 | 29.5 | 74.3 | 93.4 | 54.5 | 0.0 | 0.0 |

# A APPENDIX

## A.1 EVALUATION RESULTS OF ADVANCED MODELS

We present the evaluation results of advanced vision-language models in Table 7, including OpenAI GPT-5 [4] and o4-mini [5], Gemini-1.5-Flash Team et al. (2024), Gemini-2.5-Pro Team (2025), Gemini-3-Pro, PolyFormer Liu et al. (2023d), MDETR Kamath et al. (2021), ReLA Liu et al. (2023a), X-Decoder Zou et al. (2022), SEEM Zou et al. (2023), Gemini-1.5-Flash Team et al. (2024), Gemini-2.5-Pro Team (2025), LISA series Lai et al. (2023); Yang et al. (2023), GLaMM Rasheed et al. (2024), Llava-Next series Liu et al. (2024a), DeepSeek-VL2 series Wu et al. (2024), InternVL series Chen et al. (2024d;b); Zhu et al. (2025), Qwen2.5-VL Bai et al. (2025) series, Qwen3-VL series [6], and VLM-R1-3B Shen et al. (2025) are also included for reference.

---

[4]https://openai.com/index/introducing-gpt-5

[5]https://openai.com/index/introducing-o3-and-o4-mini

[6]https://github.com/QwenLM/Qwen3-VL

## A.2 Experimental Details

### A.2.1 Expected Output of Negative Expressions.

We mainly follow the official evaluation protocols of each model while adapting them to our benchmark. Specifically, we consider the expected outputs for negative expressions as follows:

**LLM/MLLM-based models**: These models typically possess instruction-following capabilities, such as the Qwen2.5-VL series. We define specific expected behaviors for each task: (1) Grounding task: Models should output a designated "null" bounding box: [0, 0, 0, 0]; (2) Segmentation task: Models should generate an empty mask with no pixels marked as foreground. For models with specialized output formats, we adapt our evaluation accordingly. For example, LISA uses a special segmentation token [SEG], so we employ a dual-condition check: (1) whether the output mask is empty, and (2) whether the special token is generated. If either condition indicates "no object", we consider the model's response correct for negative expressions. For grounding models that require SAM to generate segmentation masks, we consider the model's response correct if it outputs the "null" bounding box.

**Non-LLM/MLLM-based models**: We follow each model's standard output format and evaluation protocol: (1) Grounding task: These models typically output confidence scores for each bounding box. During post-processing, only bounding boxes exceeding a predefined confidence threshold are considered valid detections. For negative expressions, we expect all confidence scores to fall below this threshold. (2) Segmentation task: Models should produce empty masks. When confidence scores are available, we apply the same threshold-based evaluation: scores below the threshold indicate correct "no object" predictions.

### A.2.2 Detailed Settings

Most of the models in our evaluation are open-sourced, and we use the official implementations to conduct the experiments. For X-Decoder, SEEM models, we follow the instructions in their official repositories to load the models and run the evaluation with default hyperparameters. For the models based on the large language models, we download the models from HuggingFace and deploy them using vLLM, which speeds up the generation using some advanced techniques such as KV cache and flash attention. Under this setting, we still cost about 60 and 90 minutes to evaluate Qwen2.5-VL-7B and Qwen2.5-VL-32B models, respectively. For the generation temperature, since this value is different among the tested models, we choose the temperature from 0 to the default value and report the best results. As the evaluation is time-consuming, we only evaluate once for the non-zero temperature. We set the temperature to 0 for the models except the InternVL2.5 and InternVL3 series, where the temperature is set to 0.7.

## A.3 Additional Discussion with Related Work

While we briefly touched upon MMR, Cops-Ref, and FineCops-Ref, we present a more detailed comparison here. Although these datasets also focus on "multiple" in some aspects, they differ significantly from the diverse referring expressions from multiple perspectives of our DRef.

To the best of our knowledge, the datasets mentioned were not explicitly designed to collect multiple distinct referring expressions for a single target object. In contrast, our DRef is designed to provide *multiple diverse texts for an object from different descriptive perspectives*, which forms the foundation of our stability evaluation.

- MMR (Multi-Objects or Multi-Parts):
  - Referring expressions: MMR does not directly provide multiple descriptions for a single specific object. Instead, a single description may refer to multiple objects or multiple parts of an object. In contrast, our DRef specifies that a single object requires at least 6 non-repetitive descriptions from different perspectives, thereby providing diverse descriptions based on different visual elements in the scene.
  - Motivation: MMR aims to locate multiple entities referred to by a single expression. Our goal, however, is to evaluate model stability given multi-view descriptions for a single entity.
  - Text-object relationship: MMR represents a **one-to-many** mapping (one expression to multiple objects), whereas DRef represents a **many-to-one** mapping (multiple expressions to one object).
- FineCops-Ref (Multi Difficulty Level) In this dataset, although an object may appear in multiple descriptions within a scene as context (a reference frame), there is typically only one description that specifically targets that object. Conversely, DRef provides multiple descriptions explicitly targeting the same object, enabling the evaluation of model robustness in realistic scenarios.
- SCALAR-VG (Multi-Tasks) In SCALAR-VG, an object is associated with only one referring text. This dataset emphasizes multi-task learning. Specifically, using bounding boxes, keypoints, and polygons to localize objects, rather than providing multiple descriptions for a single object. In contrast, our DRef highlight the robustness evaluation, and provide the bounding box and segmentation mask for each labeled object.

## A.4 INSPECT QWEN2.5-VL-3B AND QWEN2.5-VL-72B

Interestingly, the Qwen2.5-VL-3B performs better on $R_{50}^{\text{Hard}}$ than the larger 7B and 32B models, achieving results comparable to the 72B model, despite having significantly lower $R_{50}^{\text{Soft}}$ and $P_{50}$. We investigate this phenomenon by: (1) visualizing and manually checking the predicted results (Figure 7 and Figure 8); and (2) conducting a quantitative analysis using the naive consistency rate.
Our analysis reveals that the 3B model tends to select one of the foreground objects and has poor comprehension of complex expressions that reference background objects (low $R_{50}^{\text{Soft}}$ and $P_{50}$). Consequently, while this simplistic selection strategy occasionally succeeds with difficult expressions that happen to reference foreground objects, it fundamentally lacks the comprehensive understanding displayed by larger models. In contrast, the 72B model demonstrates more sophisticated comprehension of both foreground and background object references, but this broader capability paradoxically makes it more susceptible to distraction from ambiguous descriptors or competing referents within complex expressions.

The following figures present: (1) the cases where Qwen-2.5-VL-3B performs better than Qwen-2.5-VL-72B, and (2) the cases where Qwen-2.5-VL-72B performs better than Qwen-2.5-VL-3B.

**Cases of Qwen-2.5-VL-3B performs better than Qwen-2.5-VL-72B** The visualized results are shown in Figure 7. The referring expressions are listed below:

1. The die with the six-dot face oriented towards the camera.

2. The third person from left to right.

3. The person closest to the individual wearing a black helmet.

4. A person wearing a helmet on head.

5. The player furthest from the scoring screen.

6. The farthest pizza from the lady.

7. The person furthest from the car.

8. The person facing toward the left side of the image from the viewer's perspective.

9. The first cup when counted from bottom to top from image viewer's perspective.

We take some subfigures as examples to analyze the behaviors of different models. In the subfigure (4) in Figure 7, the Qwen-2.5-VL-72B model is distracted from the helmet and thus fails to localize the person on the motorcycle. In the subfigure (5), the Qwen-2.5-VL-72B might be influenced by the scoring screen and fails to understand the "furthest" in the referring expression, and thus localizes the person closest to the scoring board. In subfigure (7), the Qwen-2.5-VL-72B model recognizes another in the background, but is distracted by the "car" and fails to localize the person. The referred objects are mainly distributed in the foreground, and the Qwen-2.5-VL-3B model can localize them correctly. In contrast, the Qwen-2.5-VL-72B model is prone to being distracted by other objects in the background or other non-target objects in the foreground, leading to incorrect localization.

**Cases of Qwen-2.5-VL-72B performs better than Qwen-2.5-VL-3B** The visualized results are shown in Figure 8. The referring expressions are listed below:

1. The car closest to the left edge of the image.

2. Towel on the bathtub.

3. The plate closest to the right edge of the image.

4. The sunglasses closest to the right edge of the image from the viewer's perspective.

5. The object closest to the top edge of the image.

6. The person closest to the camera that captured this image.

7. The animal closest to the upper edge of the image.

8. The car farthest from the camera that captured this image

In subfigure (2), Qwen2.5-VL-3B ignores the small towels in the image and locates the conspicuous curtain instead. In subfigure (3), the 3B model ignores the modifiers in the referring expression and always locates the central plate instead of the correct one. In subfigure (4), the 3B model fails to recognize the sunglasses and locates the fish. In subfigure (5), the 3B model can only locate the boat instead of the lifebuoy, which is relatively smaller and in the background. A similar phenomenon happens in subfigures (7) and (8), where the 3B model cannot recognize the objects in the background.

**Quantitative Analysis.** To further validate our hypothesis, we conducted a quantitative analysis using the naive consistency rate. As detailed in Section 3.3, we introduce the naive consistency rate $\hat{R}_o^{\mathrm{C}}$ of an object $o$ as the average IoU between all pairs of bounding boxes predicted for expressions referring to the same object $o$. We compute the mean naive consistency rate over the entire dataset as follows:

$$\hat{R}^{\mathrm{C}} = \frac{1}{|\mathcal{O}|} \sum_{o \in \mathcal{O}} \hat{R}_o^{\mathrm{C}}, \tag{5}$$

where $\mathcal{O}$ denotes the set of all referred objects in the benchmark. This metric evaluates the model's internal consistency in localizing the same object across varying referring expressions, independent of localization accuracy.

Table 8: **Mean naive consistency rate of Qwen2.5-VL series.**

| Metric | Qwen2.5-VL-3B | Qwen2.5-VL-7B | Qwen2.5-VL-32B | Qwen2.5-VL-72B |
|---|---|---|---|---|
| $\hat{R}^{\mathrm{C}}$ | 83.7 | 59.0 | 63.6 | 65.9 |
| $R^{\mathrm{C}}$ | 32.5 | 35.7 | 46.4 | 51.4 |
| $R_{50}^{\mathrm{Hard}}$ | 26.0 | 18.4 | 23.7 | 27.7 |
| $P_{50}$ | 37.0 | 57.8 | 63.6 | 77.4 |

As shown in Table 8, the Qwen2.5-VL-3B model achieves a significantly higher mean naive consistency rate (83.7) compared to the larger models (59.0, 63.6, and 65.9 for 7B, 32B, and 72B models, respectively). This indicates that the 3B model consistently predicts similar bounding boxes for the same object, regardless of the referring expression used.

Despite the high $\hat{R}^{\mathrm{C}}$, the 3B model's $P_{50}$ is substantially lower than that of the larger models. This suggests that the 3B model tends to weigh the linguistic constraints of the referring expressions less heavily, focusing instead on salient objects within the image. In contrast, the larger models exhibit lower consistency but higher precision. This suggests they are more responsive to specific details in the referring expressions, which results in more varied predictions for the same object as the text changes.

While we conduct analysis here using the naive mean consistency rate, it will give high score if a model consistently produces wrong results. So the adopted mean consistency rate is modulated by taking the correctness into account.

In summary, the quantitative analysis supports our earlier observations from the visualizations, confirming that the 3B model's high consistency stems from its simplistic approach of focusing on prominent foreground objects, while the larger models' lower consistency reflects they are more sensitive to the nuances in the referring expressions, increasing their likelihood of distraction.

**Conclusion** The examples above imply that the Qwen-2.5-VL-3B model mainly focuses on the foreground objects and tends to assume that one of them is the object referred to. In contrast, the Qwen-2.5-VL-72B model has a better understanding of the given image but is prone to being distracted by other objects in the background or other non-target objects in the foreground, leading to incorrect localization.

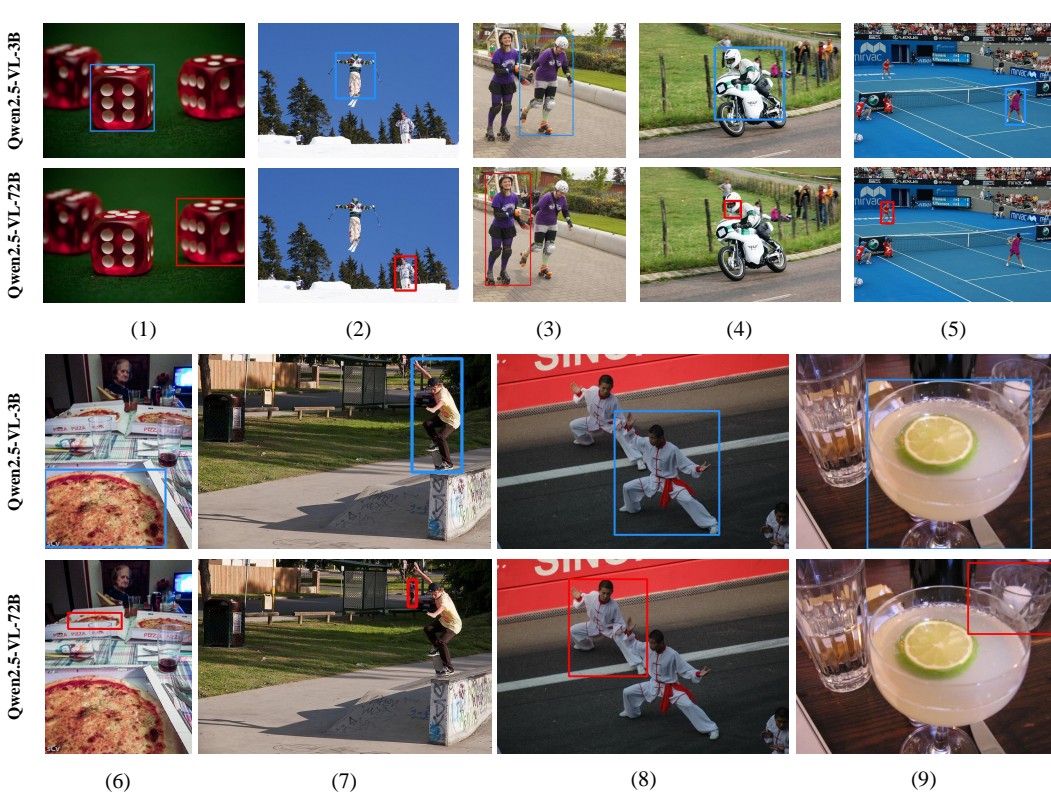

Figure 7: **Cases of 3B model better than 72B model.** The blue boxes are of Qwen2.5-VL-3B and correctly localize the referred objects, while the red boxes are of Qwen2.5-VL-72B and localize incorrect objects. Best viewed in color and zoomed in.

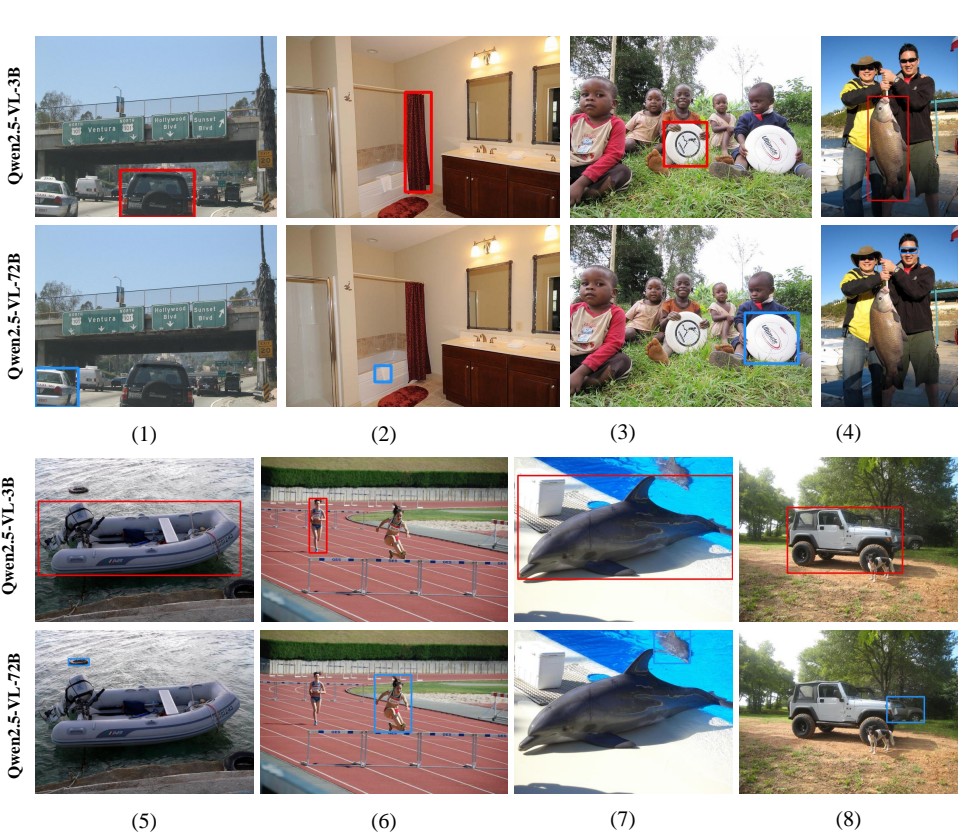

Figure 8: **Cases of 72B model better than 3B model.**

## A.5 FAILED TAGS COMBINATIONS

Using LLaVA-Next-Mistral-7B, Qwen2.5-VL-7B, Qwen2.5-VL-72B, and Qwen3-VL-32B as a representative case study, we present further analysis of the failure modes. We observed that the models *fail completely* on some tag combinations, which are listed in the Appendix.

In addition to these complete failures, we visualize the 10 combinations with the lowest $P_{50}$ in Figure 9.

Although we don't find *exactly* the same combinations that appear in each model, we find the absolute and relative position are prevalent in the majority of the failure modes. This suggests that referring expressions involving spatial relationships pose a significant challenge for such models.

Llava-Next-Mistral-7B completely failed on the following combinations of tags:

- 2D Position + 3D Position + Interaction + Possible Status
- 2D Position + 3D Position + Relative Position
- 2D Position + Attribute + Possible Status + Relative Position
- 2D Position + Possible Status + Possible Usage
- 2D Position + Possible Status + Possible Usage + Relative Position
- 2D Position + Possible Status + Reasoning
- 2D Position + Possible Usage + Reasoning + Relative Position
- 2D Position + Reasoning + Relative Position
- 2D Position + Size
- 3D Position + Interaction + Possible Status
- 3D Position + Interaction + Possible Status + Possible Usage + Relative Position
- 3D Position + Interaction + Relative Position
- Attribute + Interaction + Size
- Attribute + Possible Status + Size
- Attribute + Possible Usage + Reasoning
- Interaction + Possible Usage

Qwen2.5-VL-7B completely failed on the following combinations of tags:

- 2D Position + 3D Position + Interaction + Possible Status
- 2D Position + 3D Position + Possible Status
- 2D Position + 3D Position + Relative Position
- 2D Position + Attribute + Interaction + Possible Status + Relative Position
- 2D Position + Attribute + Possible Status + Relative Position
- 2D Position + Interaction + Relative Position
- 2D Position + Possible Status + Possible Usage
- 2D Position + Possible Status + Possible Usage + Relative Position
- 2D Position + Possible Status + Reasoning
- 2D Position + Possible Status + Reasoning + Relative Position
- 2D Position + Reasoning + Relative Position
- 3D Position + Attribute + Interaction
- 3D Position + Attribute + Interaction + Relative Position
- 3D Position + Interaction + Possible Status
- 3D Position + Interaction + Possible Status + Possible Usage + Relative Position
- 3D Position + Possible Usage + Reasoning + Relative Position

- 3D Position + Relative Position + Size
- Attribute + Interaction + Possible Usage
- Attribute + Interaction + Reasoning
- Attribute + Possible Status + Possible Usage
- Attribute + Possible Status + Size
- Interaction + Possible Usage
- Interaction + Relative Position + Size
- Reasoning + Relative Position

Qwen2.5-VL-72B completely failed on the following combinations of tags:

- 2D Position + 3D Position + Interaction + Status
- 2D Position + Interaction + Relative Position
- 2D Position + Possible Usage + Status
- 2D Position + Reasoning + Relative Position
- 2D Position + Reasoning + Status
- 2D Position + Size
- 3D Position + Interaction + Status
- Attribute + Interaction + Reasoning + Relative Position + Status
- Interaction + Possible Usage
- Interaction + Relative Position + Size
- Reasoning + Relative Position

Qwen3-VL-32B:

- 2D Position + 3D Position + Interaction + Possible Status
- 2D Position + 3D Position + Possible Status
- 2D Position + Attribute + Interaction + Possible Status + Relative Position
- 2D Position + Interaction + Relative Position
- 3D Position + Attribute + Possible Usage + Relative Position
- 3D Position + Interaction + Possible Status
- Attribute + Interaction + Possible Status + Reasoning + Relative Position
- Attribute + Interaction + Possible Usage
- Attribute + Interaction + Reasoning
- Attribute + Possible Status + Size
- Reasoning + Relative Position

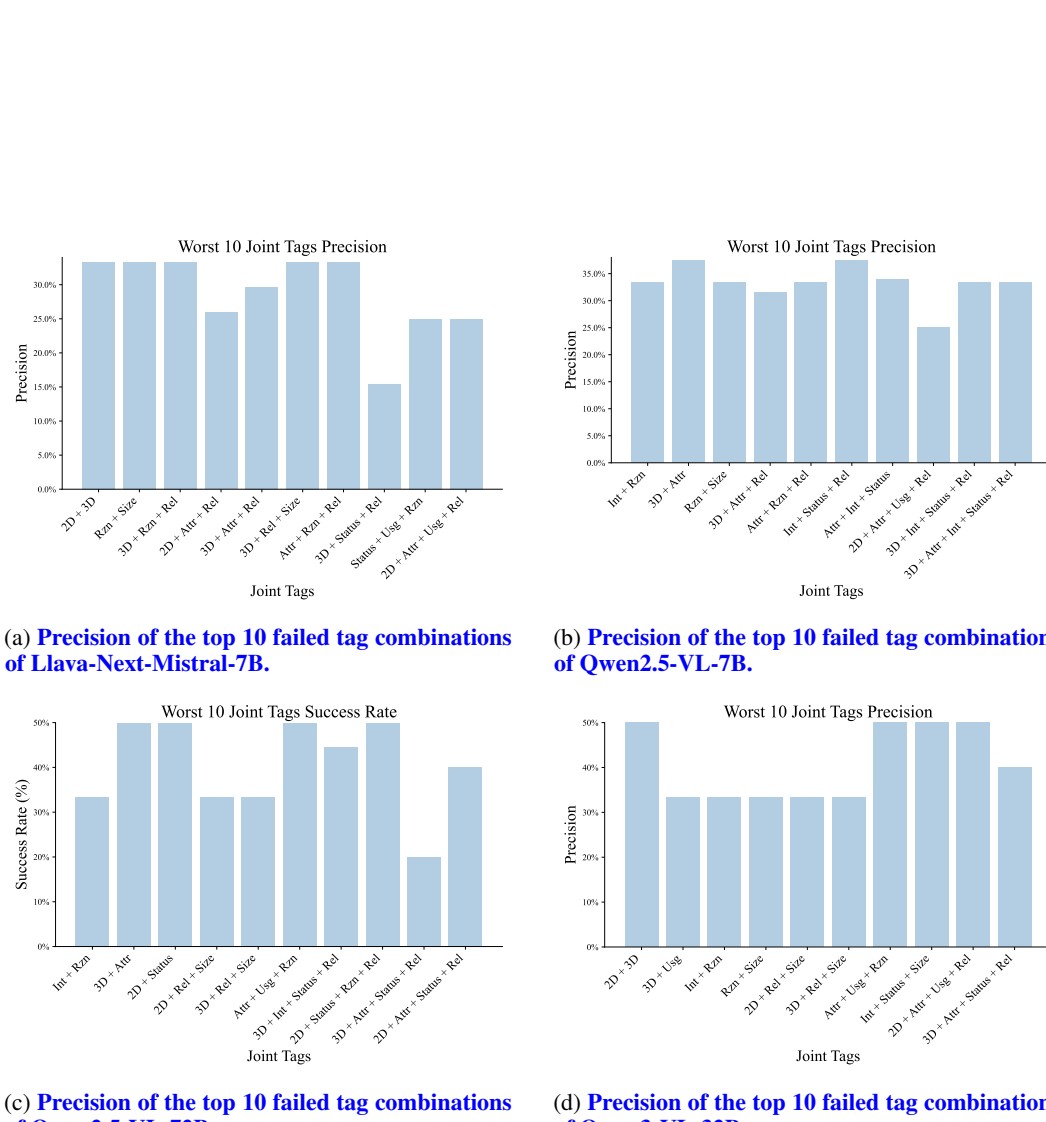

(a) **Precision of the top 10 failed tag combinations of Llava-Next-Mistral-7B.**

(b) **Precision of the top 10 failed tag combinations of Qwen2.5-VL-7B.**

(c) **Precision of the top 10 failed tag combinations of Qwen2.5-VL-72B.**

(d) **Precision of the top 10 failed tag combinations of Qwen3-VL-32B.**

Figure 9: **Precision of the top 10 failed tag combinations across different models and sizes.** The higher the better. Int: Interaction; Rzn: Reasoning; Attr: Attribute; Rel: Relative Position; Usg: Possible Usage; 2D: 2D Position; 3D: 3D Position; Status: Possible Status.

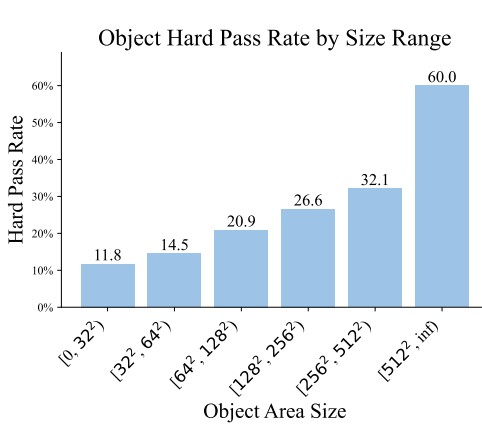

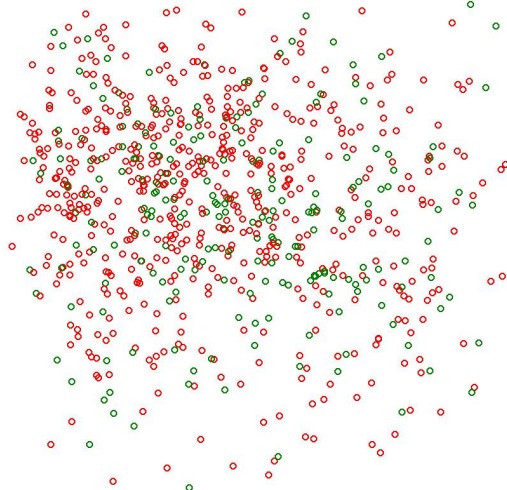

Figure 10: **Hard Pass Rate of different object sizes.** It's easier for Qwen2.5-VL-72B to succeed on larger objects.

Figure 11: **The distribution of the center of objects.** Red: failed on at least one referring expression. Green: succeeded on all referring expressions.

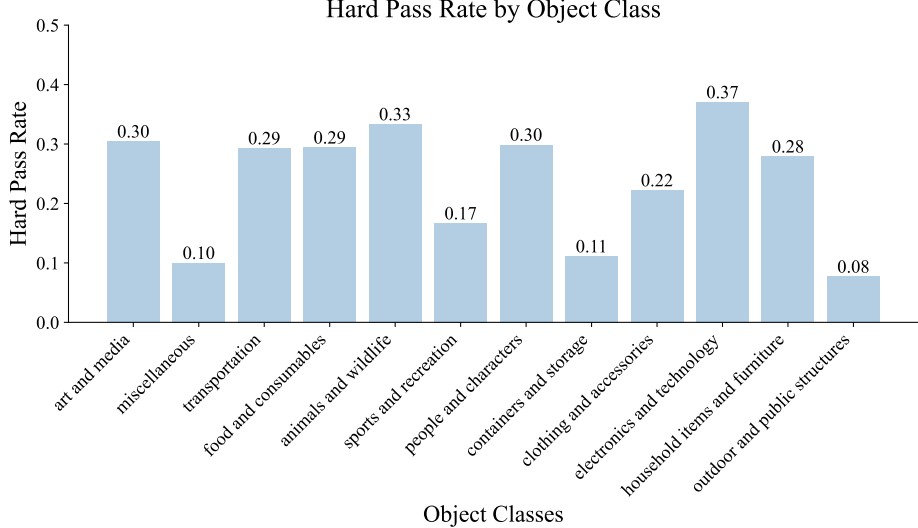

Figure 12: **Hard Pass Rate of different object classes.**

## A.6 WHAT MIGHT AFFECT MODEL PERFORMANCE?

We analyzed the impact of the following object-specific factors on Qwen2.5-VL-72B: (1) object size, (2) spatial distribution, and (3) object category.

- **Impact of Object Size:** Following the object size definitions from COCO Lin et al. (2014), we evaluated the model's hard pass rate across different sizes (Figure 10). The results indicate a positive correlation between size and performance: the model achieves a higher hard pass rate on larger objects, with performance declining as object size decreases.

- **Impact of Object Location Distribution:** Similar to DETR, we normalized object center coordinates to a uniform scale in Figure 11. The analysis demonstrates that the model's

performance is largely robust to spatial variation, showing little correlation with the normalized object locations.

- **Object Class:** Figure 12 illustrates the hard pass rate for each specific category (detailed in Figure 15). We observe that performance remains relatively consistent across the majority of categories, showing minimal variance based on object class. We observe lower scores for specific semantic categories, such as "containers and storage". We attribute this to intrinsic visual properties common to these objects, such as transparency (e.g., glass cups) and small spatial dimensions, which are known challenges in this domain.

## A.7 ANNOTATION EXAMPLES

We present some of the annotated examples in Figure 13 and Figure 14. The referring expressions are in a light-blue background, while the negative ones are in a light-gray background. The blue boxes are the ground truth of the referred objects. Segmentation masks are ignored here for simplicity.

The person closest to the lower edge of the image from viewer's perspective
The person closest to the camera that captured this image
The person carrying a blue-gray shoulder bag
The person wearing dark blue jeans
The person wearing a black hat
The person most likely to be posing for a group photo with the deer herd
The person holding a transparent umbrella in her left hand from subject's perspective
The person facing the camera

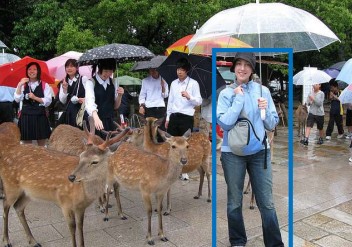

The person carrying a black backpack on her back
The person riding on a deer's back
The person holding a folded umbrella
The person holding an umbrella printed with a deer pattern

The excavator closest to the upper edge of this image
The excavator closest to the red-and-white checkered patterned object bag
The second excavator from the bottom counting upward from this image viewer's perspective
The excavator with a fully visible tail section
The excavator closest to the orange truck at the center of this image
The excavator closest to the zipper head

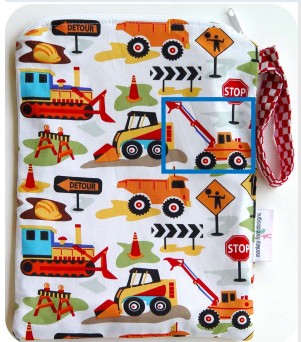

The yellow excavator
The excavator with its mechanical arm folded
The excavator with red wheels
The excavator with checkered patterns
The excavator with tracks crawler undercarriage

The vehicle that appears to be the largest
The vehicle that appears to be the tallest
The vehicle with the number '7' on its body
The vehicle that is most likely used for transporting frozen foods
The vehicle with a white compartment
The object used for transporting large quantities of goods

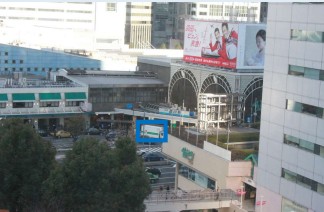

The vehicle with a person standing on its roof
The vehicle with the number '7' printed on its front section
The vehicle depicted on the billboard
The vehicle with green wheel hubs

Figure 13: Examples of DRef benchmark.

The knife closest to the right edge of the image
The farthest knife from the camera that captured this image
The knife on the food
The knife closest to the grape
The wooden knife
The knife placed on the plate closest to the bottom of the image
The knife closest to the cup in the dish
A knife that most likely to be used to spread butter

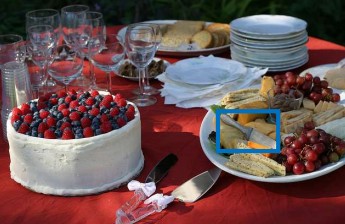

A knife cutting a cake
A knife covered with a book
A knife hanging on the wall
A knife with a red pattern

The one closest to the right edge of the image within the poster featuring a clearly visible female face
The poster featuring three people
The poster displaying the most exaggerated facial expression
The one positioned in the first row and fourth column within the poster from the viewer's perspective
The poster most likely promoting a group/team ensemble
The poster located to the right of the poster bearing the text 'CON TODOS' viewer's right
The poster positioned directly above the poster featuring the word 'LIVE'
The poster situated to the left of the poster displaying the text 'BUZONEO' viewer's left

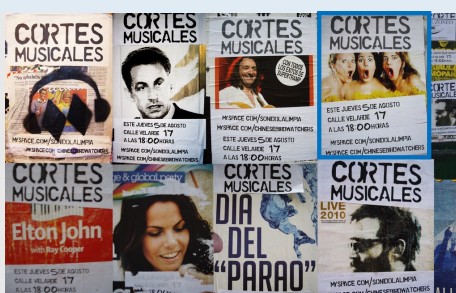

The poster with red-colored text
The poster depicting three male individuals
The blank/unprinted poster
The poster with a black background/base color
The poster containing blue-colored text

The person with white hair
The person wearing glasses
The person wearing black gloves
The person wearing red pants
The person who appears to be the oldest
The person whose own left hand is placed on another person's right leg
The person who appears to be the shortest among those seated on the sofa
The person with the highest seniority status in the family hierarchy

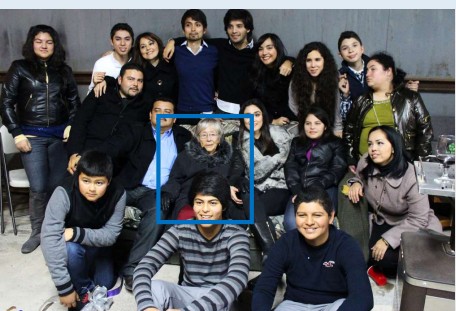

The person holding a pair of glasses
The person seated on the chair
The person wearing red clothing
The person holding a cup
The person wearing blue gloves

Figure 14: Examples of DRef benchmark.

A.8 CATEGORIES

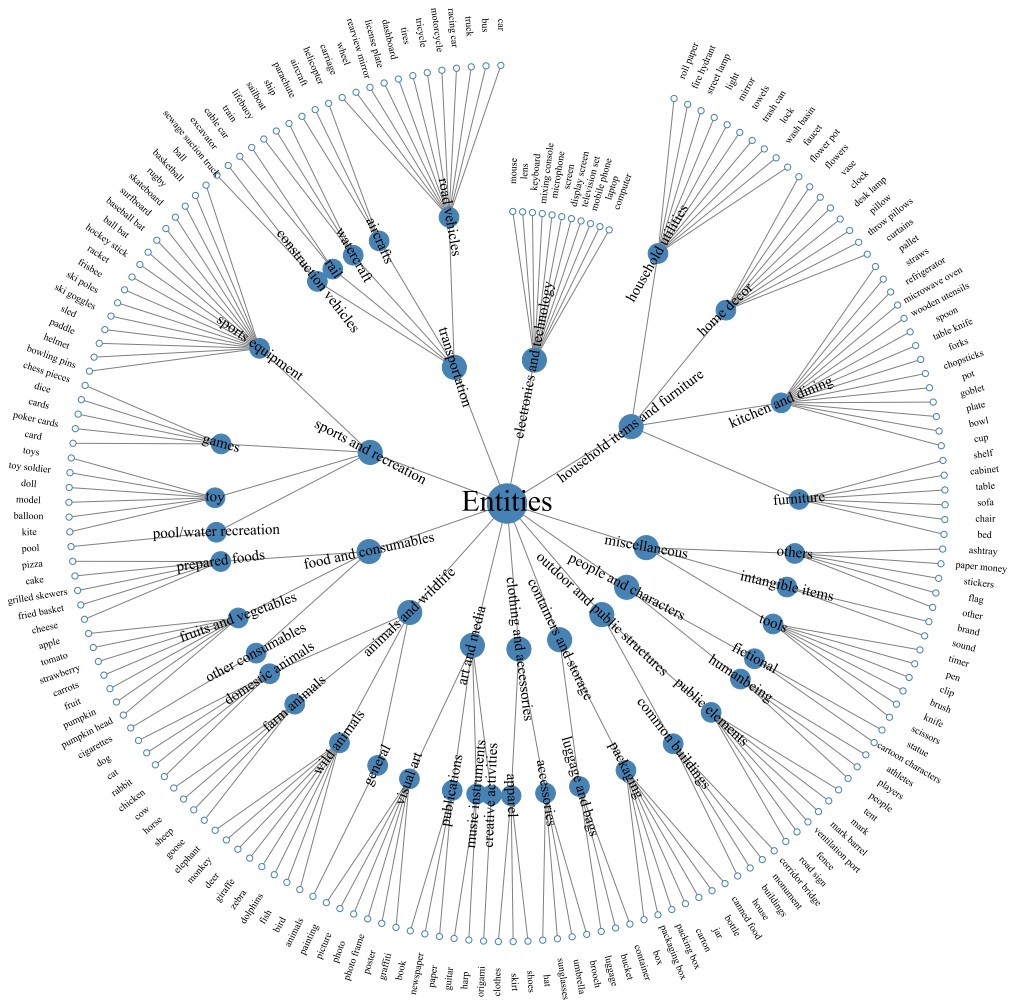

Figure 15: **The categories of annotated objects.** Parent nodes represent more generic concepts than their children.

Our annotation team annotated 824 objects in 187 categories. As illustrated in Figure 15, we organize them in a hierarchical structure, where parent nodes represent more generic concepts than their children. The statistics of the categories of the referred objects are presented in Figure 16.

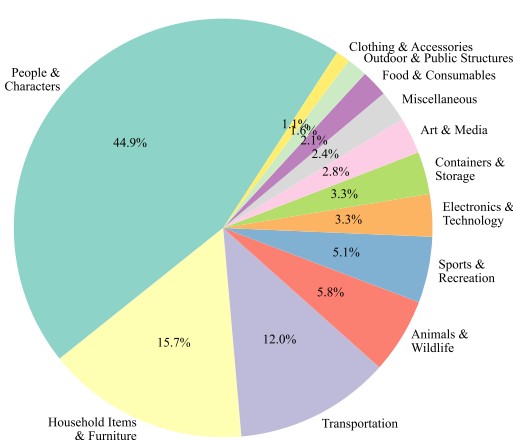

Figure 16: The statistics of the categories of the referred object.

## A.9 BROADER IMPACTS

**Potential Positive Impacts.** Our work focuses on the evaluation of the robustness of the task of referring expression comprehension of vision-language models under diverse referring expressions. We provide a benchmark with multiple diverse referring expressions for the same object. Also, the proposed metrics, Hard Pass Rate and Mean Consistency Rate, reveal that existing models are not so reliable when confronted with diverse referring expressions. Our benchmarking results encourage the community to develop more robust models to benefit real-world applications, such as autonomous driving, robotic manipulation, and human-robot interaction.

**Potential Negative Impacts.** We collect the publicly available images from the OpenImages dataset and the COCO dataset. Some of the images contain the living environment, which may include the faces of people. The private information of the people in the images may be leaked during the usage of the benchmark. A potential solution is to use AI generated images, but it's necessary to further research to ensure the generated images can be used to construct evaluation benchmarks.

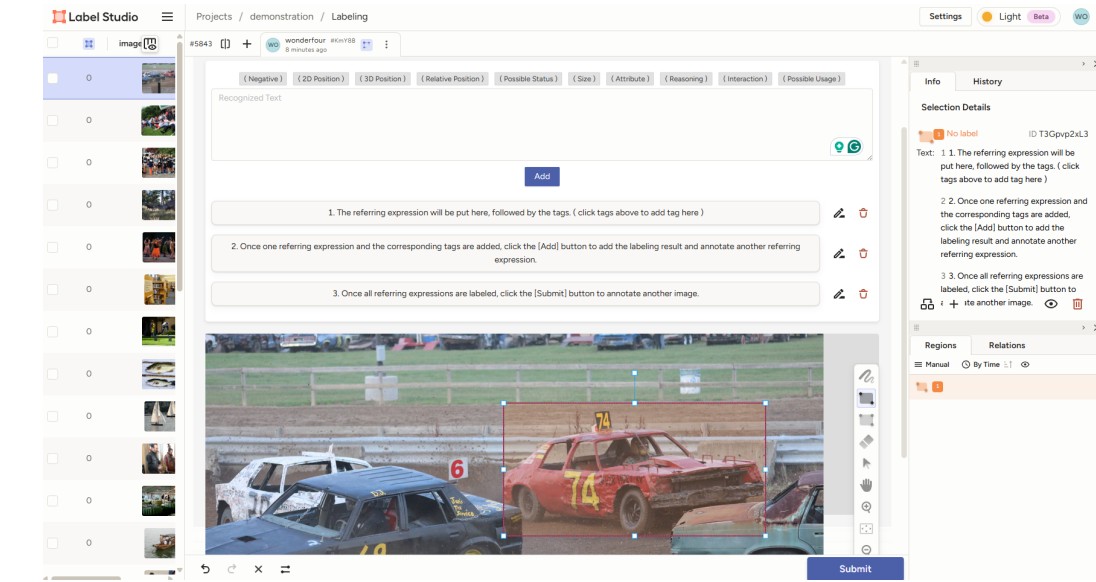

Figure 17: **The annotation interface.** The developed annotation tool based on label-studio. This tool enables the annotation of bounding boxes, segmentation masks, and referring expressions for selected objects.

A.10    MORE ANNOTATION DETAILS

We developed an annotation tool based on label-studio to annotate the bounding boxes, segmentation masks, and referring expressions for selected objects, the screenshot is shown in Figure 17. Annotators are trained to use the tool and follow the guidelines as described below.

**About the Annotators.**    The annotators are full-time employees of a labeling company, and all of them are paid according to the contract.

**Training Process.**    We trained the annotators to instruct them on how to use the annotation tool. They are told:

- Select the "Rectangle" tool to draw the bounding box of your chosen object, and adjust the box carefully to ensure it surrounds the object as closely as possible.

- Click the bounding box, then the text box will appear. Type the referring expression, and click the corresponding tags listed above the text area to attach the tags to the referring expression.

- Once one referring expression is finished, click the "Add" button to add the referring expression to the list and start a new one.

- If you find the referring expression is not correct, you can click the "Edit" button (in the shape of a pen) to correct it or the "Delete" button (in the shape of a trash) to remove it from the list.

- If you finish all the referring expressions, click the "Submit" button to submit your work.

- If you are asked to review the segmentation masks, click the "brush" tool or "Eraser" tool to adjust the segmentation masks. If the mask is totally wrong, you can click the "Delete" button and draw a new one using the "brush" tool.

- If you are asked to review the annotation results, carefully check the bounding boxes, segmentation masks, referring expressions, and the tags. If you find any mistakes, please edit them using the corresponding tools.

**Annotation Guidlines.**     The annotators are trained to follow the guidelines below:

In this annotation task, you are asked to annotate each object in the given image with multiple referring expressions from different aspects. The referring expressions should be diverse, unambiguous, and can uniquely identify the object. The annotation process is time-consuming and requires careful consideration. **You should put the quality of the annotation as the first priority.**

You should first select an object in the image with the following rules:

- If the given image is very simple, *e.g.*, the close-up of a single object, please skip it and go for another image.

- If there is only one type of object and there is not enough recognition between objects, *e.g.*, an image only contains several apples without other marks to further distinguish them, please also skip the image. Objects like dice would be OK as there are more details and they can be distinguished by the dots on the surface.

- You can select a part of the object, *e.g.*, the shirt of a person, or the wheel of a car.

- If there is only one object in the foreground, select the object in the background, or the part of the object in the foreground.

- If there are multiple objects in the foreground, you are free to select one of them or the object in the background.

You should then draw a bounding box (introduced in the tutorial of the annotation tool) around the selected object. The bounding box should be as tight as possible to the object. You can zoom in on the image for better accuracy. The bounding box would then be used to automatically generate the corresponding segmentation mask.

To describe the object from different perspectives, you can take the following table as a reference. Please carefully observe the object based on its intrinsic properties, such as color, shape, size, composition, and material, etc. Then, carefully observe the relationship and interaction between the object and its surrounding objects as well as the environment.

Table 9: The definition as well as the example of the tags in DRef.

| Tag | Description | Example |
|-----|-------------|---------|
| 2D Position | The position description on a plane, *e.g.*, image plane. | The person closest to the top of the image. |
| 3D Position | The position description on a 3D space. | The bird in the mid-air. |
| Relative Position | Describe the position of the object relative to other objects. | The person to the left of the car. |
| Size | The size of the object, typically compared with other objects. | The visually largest toy. |
| Attribute | The intrinsic properties of the object. | The red car. |
| Interaction | The interaction between the object and other objects or the environment. | The person who is speaking to the woman in red. |
| Possible Usage | The possible usage or designed purpose of the object. | The object that is used for holding garbage. |
| Possible Status | The possible status of the object. | The device that is turned on. |
| Reasoning | Requires reasoning from existing visual and textual cues. | The person most likely to be the band's lead singer. |
| Negative | The object that is not in the image. | The person who is speaking to the woman in green. |

To construct the negative referring expressions, please keep in mind that you are trying to fool a model. You should use one or more of the following strategies to modify a positive referring expression to create a negative one::

- Attribute modification: Modify one or a few key attributes of the positive expression, such as color, position, size, object categories, material properties, current status, or possible usage.

- Attribute quantity adjustment: Increase or decrease the number of specified attributes in the positive expressions.

- Fine-grained detail modification: Alter specific details within objects mentioned in the expressions. For example, "The object that is imprinted with the flag of Australia" can be modified to "The object that is imprinted with the flag of the United Kingdom."

If you are in the role of a reviewer to check the annotation results, you should carefully check the bounding boxes, segmentation masks, referring expressions, and the tags.

For the bounding boxes, please check if the bounding box is tight enough to the object. If it's not the case, please adjust the bounding box to make it meet the requirement. Don't forget to zoom in on the image for better accuracy.

For the segmentation masks, please correct the issues such as the internal holes or the inclusion of areas outside the object. Please keep in mind that the generated segmentation masks might be totally wrong, *e.g.*, there are multiple objects within the bounding box, so you are also responsible for checking if the mask belongs to the object referred to by the referring expressions. If the mask is totally wrong, please delete it and draw a new one using the "brush" tool.

For the referring expressions, please check if the referring expressions are correct and can uniquely identify the object. If it's not the case, please follow your best judgment to either edit the referring expression or delete it and write a new one. For the tags, please also check if the tags are correct based on the above table. If there are controversial tags, please remove them.

If you encounter any issues during the annotation process, please feel free to contact us for help. One of us will answer your questions as soon as possible.

