# OpenReview forum: "DRef: A Benchmark with Diverse Referring Expressions for Object Comprehension of Vision-Language Models"
_ICLR.cc/2026/Conference — Submitted to ICLR 2026_

### Official Review · Reviewer_TExx · 2025-10-24

**Soundness:** 3
**Presentation:** 3
**Contribution:** 2
**Rating:** 4
**Confidence:** 5

**Summary:**

This paper introduces DRef, a novel benchmark for evaluating the robustness of vision-language models (VLMs) in referring expression comprehension (REC). Unlike existing benchmarks that use limited, single expressions per object, DRef provides multiple diverse expressions  for the same object, along with negative expressions for non-existent objects. To rigorously evaluate model performance under this diversity, the authors propose two new metrics: Hard Pass Rate (requiring correct localization across *all* expressions for an object) and Mean Consistency Rate (measuring the stability of predictions across expressions).

**Strengths:**

1.  The motivation is well-argued. The paper clearly identifies a fundamental flaw in existing REC benchmarks — their lack of linguistic diversity — and proposes a direct, impactful solution with DRef and its new metrics.

2.  The proposed metrics (Hard Pass Rate and Mean Consistency Rate) are insightful. They provide a much more rigorous and realistic assessment of model robustness and consistency than traditional accuracy metrics, effectively exposing the limitations of current VLMs.

**Weaknesses:**

1. The claim of being the **“first systematic benchmark specifically designed to evaluate comprehensive object understanding across diverse referring expressions”** (Section 1) is potentially overstated. While DRef is highly novel, prior work like SCALAR-VG[1], FineCops-Ref [2,4], MMR [3] and  also explore multi-expression or multi-granularity reasoning, albeit not with the same focus on robustness evaluation. A more nuanced discussion of related work would strengthen the paper.

2. The **dataset size is relatively small** (824 objects, 10,963 expressions). While the quality and diversity are high, the scale is significantly smaller than benchmarks like RefCOCO (50k+ annotations). This raises questions about the statistical power of the results and the generalizability of the findings to larger, more diverse datasets.

3. The **analysis of model failure cases, while insightful, is qualitative**. The comparison between Qwen2.5-VL-3B and 72B (Appendix A.3) provides excellent examples but lacks deeper, quantitative analysis (e.g., attention map visualization, error type categorization) to explain *why* the larger model is more susceptible to distraction.

4. The **evaluation is limited to open-source models**. The results would be more impactful if they included comparisons with leading closed-source models (e.g., GPT-4V, Gemini 1.5 Pro) to provide a more complete picture of the state of the art.


[1] Yang, Xiaoyu, et al. "Enhancing visual grounding and generalization: A multi-task cycle training approach for vision-language models." arXiv 2023.

[2] Liu, Junzhuo, et al. "FineCops-Ref: A new Dataset and Task for Fine-Grained Compositional Referring Expression Comprehension." EMNLP. 2024.

[3] Jang, Donggon, et al. "MMR: A Large-scale Benchmark Dataset for Multi-target and Multi-granularity Reasoning Segmentation." ICLR 2025.

[4] Yang, Xuzheng, et al. "New Dataset and Methods for Fine-Grained Compositional Referring Expression Comprehension via Specialist-MLLM Collaboration." IEEE TPAMI 2025.

**Questions:**

See Weakness

---

> ### Author Response · Authors · 2025-11-27
>
> Thank you for reviewing our paper and for your valuable feedback. Below, we address your concerns point by point and we’ve revised our paper according to your suggestions. We would appreciate it if you could let us know whether your concerns are addressed by our response.

---

> > ### Author Response · Authors · 2025-11-27
> > **Q1. Related Work**
> >
> > > While DRef is highly novel, prior work like SCALAR-VG[1], FineCops-Ref [2,4], MMR [3] and also explore multi-expression or multi-granularity reasoning, albeit not with the same focus on robustness evaluation. A more nuanced discussion of related work would strengthen the paper.
> >
> > While we briefly touched upon MMR, Cops-Ref, and FineCops-Ref (line 153–158), we present a more detailed comparison including SCALAR-VG here, and we have updated the manuscript accordingly (Appendix A.3).
> >
> > To the best of our knowledge, the datasets mentioned were not explicitly designed to collect multiple distinct referring expressions for a single target object. In contrast, our DRef is designed to provide _multiple diverse texts for an object from different descriptive perspectives_, which forms the foundation of our stability evaluation. Although these datasets address certain aspects of multiplicity, they fundamentally differ from **DRef**, which is uniquely characterized by diverse referring expressions grounded in multiple perspectives.
> >
> > - **MMR (Multi-Objects or Multi-Parts):**
> >     - **Referring expressions**: MMR does not directly provide multiple descriptions for a single specific object. Instead, a single description may refer to multiple objects or multiple parts of an object. In contrast, our **DRef** specifies that a single object requires at least 6 non-repetitive descriptions from different perspectives, thereby providing diverse descriptions based on different visual elements in the scene.
> >     - **Motivation**: MMR aims to locate multiple entities referred to by a single expression. Our goal, however, is to evaluate model stability given multi-view descriptions for a single entity.
> >     - **Text-object relationship**: MMR represents a **one-to-many** mapping (one expression to multiple objects), whereas DRef represents a **many-to-one** mapping (multiple expressions to one object).
> > - **FineCops-Ref (Multi Difficulty Level)** In this dataset, although an object may appear in multiple descriptions within a scene as context (a reference frame), there is typically only one description that specifically targets that object. Conversely, **DRef** provides multiple descriptions explicitly targeting the same object, enabling the evaluation of model robustness in realistic scenarios.
> > - **SCALAR-VG (Multi-Tasks)** In SCALAR-VG, an object is associated with only one referring text. This dataset emphasizes multi-task learning. Specifically, using bounding boxes, keypoints, and polygons to localize objects, rather than providing multiple descriptions for a single object. In contrast, our DRef highlight the robustness evaluation, and provide the bounding box and segmentation mask for each labeled object.

---

> > ### Author Response · Authors · 2025-11-27
> > **Q2. Dataset Scale**
> >
> > > The **dataset size is relatively small** (824 objects, 10,963 expressions). While the quality and diversity are high, the scale is significantly smaller than benchmarks like RefCOCO (50k+ annotations).
> >
> > **DRef already Exposes Critical Robustness Issue.** DRef is specifically designed to evaluate model robustness against diverse referring expressions. Our empirical results highlight a critical gap: existing comprehension models exhibit significant performance degradation when presented with such diverse referring expressions.
> >
> > **Enhanced Complexity of DRef.** DRef presents a significant challenge and serves as a strong extension to existing *evaluation benchmarks*. To ensure a fair comparison, we conduct a comparative analysis against the evaluation subsets (val + testA + testB) of RefCOCO. We summarize the comparison in the table below:
> >
> > - **Scale:** DRef contains approximately 11k referring expressions, which is comparable in scale to the validation subset of RefCOCO.
> > - **Visual Diversity:** Images in DRef are sourced from two distinct datasets (COCO val 2017 and OpenImages test v7), whereas RefCOCO relies exclusively on COCO val 2014. Consequently, DRef offers greater visual domain diversity.
> > - **Difficulty Gap:** Our DRef poses a significantly greater challenge to existing grounding models (see Table 5, page 9 in our paper). On DRef, model performance ranges from 48.3 to 70.8 mIoU, whereas on val and testA subset of RefCOCO, models achieve scores exceeding 90 mIoU.
> > - **Robustness Testing:** DRef exposes robustness issues ($R^{\text{hard}}_{50}$) in current frontier models by including diverse and challenging referring expressions for the same target object, which is not available in RefCOCO.
> >
> > **Table Q2-1.** Comparison between our DRef and evaluation subsets of RefCOCO.
> >
> > |                                           | DRef (Ours)                                                          | RefCOCO (val/testA/testB)                            |
> > | ----------------------------------------- | -------------------------------------------------------------------- | ---------------------------------------------------- |
> > | Number of Referring Expressions           | ~11k                                                                 | ~11k / ~5k / ~5k                                         |
> > | Image Source(s)                           | COCO val 2017 & OpenImages test v7 (release date: 25/10/2022 [1]); | COCO val 2014;                                       |
> > | Difficulty (Table 4, page 9 in the paper) | Hard: grounding mIoU ranges from 48.3 to 70.8;                       | Easy: most grounding mIoU scores are higher than 90; |
> > | Robustness Evaluation                     | Yes (enabled by on average 8.3 expressions per object);              | No;                                                  |
> >
> > **Comparison with Recent Multimodal Benchmarks.** As shown in the table below, several recent challenging benchmarks in multimodal research contain between 300 and 1,500 images. These benchmarks evaluate multimodal capabilities—including visual perception, language comprehension, and visual reasoning—via tasks such as QA or segmentation. Similarly, our DRef benchmark assesses perception and comprehension through object localization and segmentation tasks that require complex reasoning (spatial, logical, and commonsense).
> >
> > **Table Q2-2.** Comparison between our DRef and other recent multimodal benchmarks.
> >
> > | Benchmark    | Venue      | #images              | #questions           |
> > | ------------ | ---------- | -------------------- | -------------------- |
> > | MMVP[1]      | CVPR'24    | 300                  | 300                  |
> > | MMVet[2]     | ICML'24    | 200                  | 218                  |
> > | MMStar[3]    | NeurIPS'24 | 1500                 | 1500                 |
> > | ReasonSeg[4] | CVPR'24    | 200(val) + 779(test) | 200(val) + 779(test) |
> > | DRef (Ours)  | ---        | 824                  | 10963                |
> >
> > [1] https://storage.googleapis.com/openimages/web/news.html
> >
> > [2] Tong, S., Liu, Z., Zhai, Y., Ma, Y., LeCun, Y., & Xie, S. (2024). Eyes wide shut? exploring the visual shortcomings of multimodal llms. In _Proceedings of the IEEE/CVF Conference on Computer Vision and Pattern Recognition_ (pp. 9568-9578).
> >
> > [3] Yu, W., Yang, Z., Li, L., Wang, J., Lin, K., Liu, Z., ... & Wang, L. MM-Vet: Evaluating Large Multimodal Models for Integrated Capabilities. In _Forty-first International Conference on Machine Learning_.
> >
> > [4] Chen, L., Li, J., Dong, X., Zhang, P., Zang, Y., Chen, Z., ... & Zhao, F. (2024). Are we on the right way for evaluating large vision-language models?. _Advances in Neural Information Processing Systems_, _37_, 27056-27087.
> >
> > [5] Lai, X., Tian, Z., Chen, Y., Li, Y., Yuan, Y., Liu, S., & Jia, J. (2024). Lisa: Reasoning segmentation via large language model. In _Proceedings of the IEEE/CVF Conference on Computer Vision and Pattern Recognition_ (pp. 9579-9589).

---

> > ### Author Response · Authors · 2025-11-27
> > **Q3. Quantitative Analysis**
> >
> > > The **analysis of model failure cases, while insightful, is qualitative.** The comparison between Qwen2.5-VL-3B and 72B (Appendix A.3) provides excellent examples but lacks deeper, quantitative analysis (e.g., attention map visualization, error type categorization) to explain _why_ the larger model is more susceptible to distraction.
> >
> > We use a naive consistency rate (different from consistency rate in Equation 4) to present the quantitative analysis. We have revised the manuscript and put the details in Appendix (Section A4, line 1084 to 1085 in page 21 and line 1142 to 1170 in page 22). **Please note that naive consistency rate is different from the consistency adopted in our manuscript.**
> >
> > **Define Mean Naive Consistency Rate.** In Section 3.3, we introduce the naive consistency rate $\hat{R}^{\text{C}}_{o}$ (Equation 3) of an object $o$ as the average IoU between all pairs of bounding boxes predicted for expressions referring to the same object $o$. Here, we calculate the mean naive consistency rate over all objects:
> >
> > $$
> > \hat{R}^{\text{C}} = \frac{1}{|\mathcal{O}|} \sum_{o \in \mathcal{O}} \hat{R}^{\text{C}}_{o},
> > $$
> >
> > where $\mathcal{O}$ is the set of all referred objects in the benchmark. This metric reflects the model's overall consistency in localizing the same object across different referring expressions, regardless of correctness.
> >
> > **Table Q3: Mean naive consistency rate of Qwen2.5-VL series.**
> >
> > | Metric                               | Qwen2.5-VL-3B | Qwen2.5-VL-7B | Qwen2.5-VL-32B | Qwen2.5-VL-72B |
> > | :----------------------------------- | :-----------: | :-----------: | :------------: | :------------: |
> > | $\hat{R}^{\text{C}}$ (Defined Above) |     83.7      |     59.0      |      63.6      |      65.9      |
> > | ${R}^{\text{C}}$ (Adopted in Paper)  |     32.5      |     35.7      |      46.4      |      51.4      |
> > | $R^{\text{hard}}_{{50}}$             |     26.0      |     18.4      |      23.7      |      27.7      |
> > | $P_{50}$                             |     37.0      |     57.8      |      63.6      |      77.4      |
> >
> > **Analysis.** As shown in the table above, the Qwen2.5-VL-3B model achieves a significantly higher mean naive consistency rate (83.7) compared to the larger models (59.0, 63.6, and 65.9 for 7B, 32B, and 72B models, respectively). This indicates that the 3B model consistently predicts similar bounding boxes for the same object, regardless of the referring expression used.
> >
> > Despite the high $\hat{R}^{\text{C}}$, the 3B model's $P_{50}$ is substantially lower than that of the larger models. This suggests that the 3B model tends to weigh the linguistic constraints of the referring expressions less heavily, focusing instead on salient objects within the image. In contrast, the larger models exhibit lower consistency but higher precision. This suggests they are more responsive to specific details in the referring expressions, which results in more varied predictions for the same object as the text changes.
> >
> > While we conduct analysis here using the naive mean consistency rate, as we discussed in the manuscript (line 354-355, page 8), it will give high score if a model consistently produces wrong results. So the adopted mean consistency rate is modulated by taking the correctness into account.
> >
> > In summary, the quantitative analysis supports our earlier observations from the visualizations, confirming that the 3B model's high consistency stems from its simplistic approach of focusing on prominent foreground objects, while the larger models' lower consistency reflects they are more sensitive to the nuances in the referring expressions, increasing their likelihood of distraction.

---

> > ### Author Response · Authors · 2025-11-27
> > **Q4. Additional Frontier Models**
> >
> > > The **evaluation is limited to open-source models**. The results would be more impactful if they included comparisons with leading closed-source models (e.g., GPT-4V, Gemini 1.5 Pro) to provide a more complete picture of the state of the art.
> >
> > In the original Appendix, we provided results for Gemini-1.5-Flash. To ensure a more comprehensive assessment, we have expanded our evaluation to include other advanced models here (Gemini series, Qwen3-VL series, OpenAI GPT-5, and OpenAI o4-mini), highlighting the performance limitations of these systems. We have revised the paper and put the results and discussion in Section 4.2 (Table 3 in page 8, line 427 page 8 to line 452 page 9).
> >
> > **Table Q4**: Grounding results of frontier multi-modal models.
> >
> > | Models         | Pos $R^{\text{hard}}_{50}$ | Pos $P_{50}$ | Pos $R^{\text{soft}}_{50}$ | Pos $R^{C}$ | Neg $R^{\text{hard}}$ | Neg $P$ |
> > | -------------- | -------------------------- | ------------ | -------------------------- | ----------- | -------------------------- | ------------ |
> > | Gemini-2.5-Pro | 3.2                        | 6.6          | 9.6                        | 12.3        | 1.3                        | 26.7         |
> > | Gemini-3-Pro   | 3.3                        | 6.7          | 7.8                        | 12.8        | 0.4                        | 21.3         |
> > | OpenAI GPT-5   | 0.7                        | 9.2          | 30.0                       | 9.0         | 1.1                        | 24.8         |
> > | OpenAI o4-mini | 2.5                        | 27.1         | 56.8                       | 13.3        | 6.8                        | 53.0         |
> > | Qwen3-VL-2B    | 17.2                       | 51.8         | 80.3                       | 32.5        | 0.0                        | 0.0          |
> > | Qwen3-VL-4B    | 19.5                       | 67.8         | 94.2                       | 41.9        | 0.0                        | 3.2          |
> > | Qwen3-VL-8B    | 18.8                       | 67.5         | 95.0                       | 40.9        | 0.0                        | 1.1          |
> > | Qwen3-VL-32B   | 32.8                       | 78.1         | 97.5                       | 53.8        | 0.5                        | 21.1         |
> >
> > We have integrated these results into Section 4.2 to strengthen the experimental analysis. Furthermore, we included the a discussion regarding these cutting-edge closed-source models here:
> >
> > **Discussion on the results of advanced closed-source multi-modal models.** We observe that GPT-5, o4-mini, Gemini-2.5-Pro and Gemini-3-Pro yield poor results. We suspect that these frontier closed-source models have not been specifically optimized for grounding tasks. To the best of our knowledge, **there is almost no existing literature directly employing** these frontier models for grounding tasks. Furthermore, neither the technical report of Gemini-2.5-Pro [1] nor the model cards of GPT-5 and o4-mini [3,4] include evaluations on visual grounding tasks to demonstrate their capabilities in this domain. While the Qwen2.5-VL technical report [2] includes benchmarks using Gemini-1.5-Pro, the results indicate that Gemini-1.5-pro significantly underperforms compared to other models on grounding tasks, which serves as further evidence that these frontier models lack specific optimization for visual grounding.
> >
> > [1] Comanici, Gheorghe, Eric Bieber, Mike Schaekermann, Ice Pasupat, Noveen Sachdeva, Inderjit Dhillon, Marcel Blistein et al. "Gemini 2.5: Pushing the frontier with advanced reasoning, multimodality, long context, and next generation agentic capabilities." _arXiv preprint arXiv:2507.06261_ (2025).
> >
> > [2] Bai, Shuai, Keqin Chen, Xuejing Liu, Jialin Wang, Wenbin Ge, Sibo Song, Kai Dang et al. "Qwen2. 5-vl technical report." _arXiv preprint arXiv:2502.13923_ (2025).
> >
> > [3] https://openai.com/index/introducing-gpt-5/
> >
> > [4] https://openai.com/index/introducing-o3-and-o4-mini/

---

### Official Review · Reviewer_1Wzo · 2025-10-26

**Soundness:** 3
**Presentation:** 3
**Contribution:** 2
**Rating:** 4
**Confidence:** 4

**Summary:**

The authors introduce DRef, a new Referring Expression Comprehension (REC) benchmark that contains 10963 manually annotated expressions for 824 objects (187 categories) drawn from COCO-val-2017 and OpenImages-V7-test. Two new metrics are proposed:
Hard-Pass Rate (R^Hard): an object is counted correct only if all its expressions are localized (IoU ≥ 0.5).
Mean Consistency Rate (R^C): average pair-wise IoU among predicted boxes for different expressions of the same object, weighted by mean IoU to ground-truth. Experiments on some VLMs show a large gap between the conventional “single-expression” accuracy and R^Hard, revealing that current systems are far less robust to linguistic variation than previously thought.

**Strengths:**

1. REC benchmark that explicitly ties multiple linguistically diverse expressions to the same object instance and enforces hard consensus at the object level.
2. Careful two-pass annotation protocol, ambiguity checks, hierarchical tag set (9 positive + 1 negative), public release of data + code.
3. R^Hard and R^C quantify complementary aspects and expose failure modes that single-expression mIoU hides.

**Weaknesses:**

1. The scalability of the dataset is small, e.g., 824 objects is small compared with > 50k in RefCOCO; only two image sources.

2. This paper only gives a conclusion and a benchmark that existing methods are far less robust to linguistic variation than previously thought. However, no solution is presented, which limits the contribution of this paper. In addition, only the evaluation data is provided. Even though we are aware of this conclusion, what can the community do to address such a problem?

**Questions:**

Refer to Weaknesses.

---

> ### Author Response · Authors · 2025-11-27
>
> Thank you for your constructive comments and suggestions. We have revised our paper according to your comments. We respond to your questions below and would appreciate it if you could let us know if our response addresses your concerns.

---

> > ### Author Response · Authors · 2025-11-27
> > **Q1. Dataset Scale**
> >
> > > The scalability of the dataset is small, e.g., 824 objects is small compared with > 50k in RefCOCO; only two image sources.
> >
> > **DRef already Exposes Critical Robustness Issue.** DRef is specifically designed to evaluate model robustness against diverse referring expressions. Our empirical results highlight a critical gap: existing comprehension models exhibit significant performance degradation when presented with such diverse referring expressions.
> >
> > **Enhanced Complexity of DRef.** DRef presents a significant challenge and serves as a strong extension to existing *evaluation benchmarks*. To ensure a fair comparison, we conduct a comparative analysis against the evaluation subsets (val + testA + testB) of RefCOCO. We summarize the comparison in the table below:
> >
> > - **Scale:** DRef contains approximately 11k referring expressions, which is comparable in scale to the validation subset of RefCOCO.
> > - **Visual Diversity:** Images in DRef are sourced from two distinct datasets (COCO val 2017 and OpenImages test v7), whereas RefCOCO relies exclusively on COCO val 2014. Consequently, DRef offers greater visual domain diversity.
> > - **Difficulty Gap:** Our DRef poses a significantly greater challenge to existing grounding models (see Table 5, page 9 in our paper). On DRef, model performance ranges from 48.3 to 70.8 mIoU, whereas on val and testA subset of RefCOCO, models achieve scores exceeding 90 mIoU.
> > - **Robustness Testing:** DRef exposes robustness issues ($R^{\text{hard}}_{50}$) in current frontier models by including diverse and challenging referring expressions for the same target object, which is not available in RefCOCO.
> >
> > **Table Q1-1.** Comparison between our DRef and evaluation subsets of RefCOCO.
> >
> > |                                           | DRef (Ours)                                                          | RefCOCO (val/testA/testB)                            |
> > | ----------------------------------------- | -------------------------------------------------------------------- | ---------------------------------------------------- |
> > | Number of Referring Expressions           | ~11k                                                                 |  ~11k / ~5k / ~5k                                         |
> > | Image Source(s)                           | COCO val 2017 & OpenImages test v7 (release date: 25/10/2022 [1]); | COCO val 2014;                                       |
> > | Difficulty (Table 5, page 9 in the paper) | Hard: grounding mIoU ranges from 48.3 to 70.8;                       | Easy: most grounding mIoU scores are higher than 90; |
> > | Robustness Evaluation                     | Yes (enabled by on average 8.3 expressions per object);              | No;                                                  |
> >
> > **Comparison with Recent Multimodal Benchmarks.** As shown in the table below, several recent challenging benchmarks in multimodal research contain between 300 and 1500 images. These benchmarks evaluate multimodal capabilities—including visual perception, language comprehension, and visual reasoning—via tasks such as QA or segmentation. Similarly, our DRef benchmark assesses perception and comprehension through object localization and segmentation tasks that require complex reasoning (spatial, logical, and commonsense).
> >
> > **Table Q1-2.** Comparison between our DRef and other recent multimodal benchmarks.
> >
> > | Benchmark    | Venue      | #images              | #questions           |
> > | ------------ | ---------- | -------------------- | -------------------- |
> > | MMVP[2]      | CVPR'24    | 300                  | 300                  |
> > | MMVet[3]     | ICML'24    | 200                  | 218                  |
> > | MMStar[4]    | NeurIPS'24 | 1500                 | 1500                 |
> > | ReasonSeg[5] | CVPR'24    | 200(val) + 779(test) | 200(val) + 779(test) |
> > | DRef (Ours)  | ---        | 824                  | 10963                |
> >
> > [1] https://storage.googleapis.com/openimages/web/news.html
> >
> > [2] Tong, S., Liu, Z., Zhai, Y., Ma, Y., LeCun, Y., & Xie, S. (2024). Eyes wide shut? exploring the visual shortcomings of multimodal llms. In _Proceedings of the IEEE/CVF Conference on Computer Vision and Pattern Recognition_ (pp. 9568-9578).
> >
> > [3] Yu, W., Yang, Z., Li, L., Wang, J., Lin, K., Liu, Z., ... & Wang, L. MM-Vet: Evaluating Large Multimodal Models for Integrated Capabilities. In _Forty-first International Conference on Machine Learning_.
> >
> > [4] Chen, L., Li, J., Dong, X., Zhang, P., Zang, Y., Chen, Z., ... & Zhao, F. (2024). Are we on the right way for evaluating large vision-language models?. _Advances in Neural Information Processing Systems_, _37_, 27056-27087.
> >
> > [5] Lai, X., Tian, Z., Chen, Y., Li, Y., Yuan, Y., Liu, S., & Jia, J. (2024). Lisa: Reasoning segmentation via large language model. In _Proceedings of the IEEE/CVF Conference on Computer Vision and Pattern Recognition_ (pp. 9579-9589).

---

> > ### Author Response · Authors · 2025-11-27
> > **Q2. Potential Solution**
> >
> > > Even though we are aware of this conclusion, what can the community do to address such a problem?
> >
> > **Our Conclusion:** Our preliminary experiments suggest that **Reinforcement Fine-Tuning (RFT)** is a promising direction for improving the robustness of REC models.
> >
> > **Our Solution:** We conducted a two-stage fine-tuning process on Qwen2.5-VL-7B using 368 **additionally labeled objects** in extra images. Specifically, we applied Supervised Fine-Tuning (SFT) followed by RFT with GRPO on the same dataset. We denote this model as Qwen2.5-VL-7B-SFT-RFT. We have revised the paper and put the details and results to Section 4.1 (line 375 to 377, page 7), Section 4.2 (Table 3, page 8; line 427 page 8 to 451 page 9).
> >
> > **Table Q2.** Results of different fine-tuning policy for Qwen2.5-VL-7B.
> >
> > | Models                   | $R^{\text{hard}}_{50}$ | $P_{50}$ | $R^{\text{soft}}_{50}$ | $R^{C}$ |
> > | ------------------------ | ---------------------- | -------- | ---------------------- | ------- |
> > | Qwen2.5-VL-7B (baseline) | 18.4                   | 57.8     | 88.0                   | 35.7    |
> > | Qwen2.5-VL-7B-SFT        | 16.9                   | 66.3     | 94.2                   | 39.8    |
> > | $\Delta$                 | -1.5                   | +8.5     | +6.2                   | +4.1    |
> > | Qwen2.5-VL-7B-RFT        | 24.5                   | 74.7     | 95.8                   | 50.5    |
> > | $\Delta$                 | +6.1                   | +16.9    | +7.8                   | +14.8   |
> > | Qwen2.5-VL-7B-SFT-RFT    | 30.0                   | 74.7     | 94.1                   | 52.2    |
> > | $\Delta$                 | +11.6                  | +16.9    | +6.1                   | +16.5   |
> >
> > **Analysis:** Our Qwen2.5-VL-7B-SFT-RFT model achieves 30.0 on $R_{50}^{\text{hard}}$, significantly outperforming the baseline Qwen2.5-VL-7B (18.4) and even surpassing the much larger Qwen2.5-VL-72B (27.7). While the SFT stage enhances $P_{50}$​ (+8.5) and $R^C$(+4.1), it leads to a decline in $R_{50}^{\text{hard}}$(-1.5). This indicates a trade-off where the model learns something new but experiences some forgetting on challenging samples. Directly applying RFT with GRPO improves $P_{50}$​ (+16.9), $R^C$ (+14.8), and $R_{50}^{\text{hard}}$ (+6.1), showing it can improve the robustness of the model. Finally, adopting the two-stage fine-tuning process further improves the baseline on $P_{50}$​ (+16.9), $R^C$ (+16.5), and $R_{50}^{\text{hard}}$ (+11.6), indicating the initial SFT stage is important. We think SFT helps model learning to capture more details of an object.
> >
> > In a word, these results indicate that RFT is a potential solution for enhancing the robustness of VLMs against diverse referring expressions. Furthermore, as current RFT research predominantly targets domains such as mathematics and coding, we believe the underlying mechanisms and key factors of RFT in the context of robust visual grounding warrant further investigation.

---

### Official Review · Reviewer_9YEC · 2025-10-28

**Soundness:** 3
**Presentation:** 3
**Contribution:** 3
**Rating:** 6
**Confidence:** 4

**Summary:**

Referring expression comprehension (REC) is a fundamental vision-language task focused on localizing specific objects based on natural language descriptions. However, existing REC benchmarks typically provide only a single referring expression per target object, making it difficult to assess the robustness of vision–language models (VLMs) to variations in semantic descriptions. To address this limitation, this paper introduces a new benchmark, Diverse Referring Expressions for Object Comprehension (DRef), in which each object is associated with multiple distinct referring expressions—on average 8.3 positive and 5 negative expressions per object. To evaluate model robustness under linguistic diversity, the authors propose two complementary metrics: (1) Hard Pass Rate, which measures the ability to consistently localize the correct object across all valid expressions, and (2) Mean Consistency Rate, which quantifies the prediction consistency when the referring expression changes. Extensive experiments conducted on this benchmark reveal that current state-of-the-art VLMs exhibit notable limitations in handling diverse referring expressions.

**Strengths:**

Traditional REC benchmarks were proposed before the advancement happened in MLLM; the expression diversity is limited overall. Besides, there is no existing benchmark trying to test whether the VLMs can successfully locate the same object regardless of the referring expressions used. This dataset positions itself well to complement the current REC benchmarks. The proposed benchmark is also of relatively high quality. 1) It contains different referring expression types such as: position/ size/attribute/ interaction and negative objects. 2)  box and human corrected mask annotation are provided to facilitate the evaluation on both the detection and segmentation sides. 3) Quality control process is implemented to reduce the ambiguity and ensure the dataset quality.

Based on the new proposal evaluation sets, this paper also proposes two complementary metrics, hard pass rate and mean consistency rate. Both metrics aim to quantify the robustness of the VLM to different referring expressions from different perspectives; they reflect an interesting new assessment of the model capacity.  Comprehensive benchmarking is conducted on different models.

**Weaknesses:**

I appreciate the group-wise performance analysis based on the DRef tag space shown in Figure 6. The results clearly indicate that larger models exhibit improved robustness to variations in referring expressions. However, since only the top 10 most frequent tag combinations are presented, it would be informative to also identify which combinations lead to the most significant failures across models. For instance, Qwen2.5-VL-72B achieves a P50 of 77.4, yet its R50-HARD drops to roughly 27.7, suggesting that certain linguistic patterns or tag combinations pose substantial difficulties even for the strongest systems. Additional analysis on these failure modes would strengthen the insights.

It would also be valuable to examine whether robustness to expression diversity correlates with object-specific factors, such as object size, category, or higher-level semantic grouping. Such analysis could help clarify whether failures arise primarily from language ambiguity or from visual complexity.

The manuscript states that the benchmark contains 824 objects, and the appendix notes 824 images, implying that each image contains exactly one annotated object. If this is indeed the intended dataset design, it would be helpful to emphasize this point clearly in the main text, as readers might otherwise question. Furthermore, the authors may wish to discuss how the evaluation framework generalizes to scenarios involving multiple target objects within the same scene, and whether current models show differing behavior when handling single-target versus multi-target referring expressions.

**Questions:**

See above

---

> ### Author Response · Authors · 2025-11-27
>
> Thank you for reviewing our paper and for your valuable feedback. Below, we address your concerns point by point and we’ve revised our paper according to your suggestions. We would appreciate it if you could let us know whether your concerns are addressed by our response.

---

> > ### Author Response · Authors · 2025-11-27
> > **Q1. Analysis on the Failed Cases**
> >
> > > it would be informative to also identify which combinations lead to the most significant failures across models
> >
> > **Analysis Setup.** Using LLaVA-Next-Mistral-7B, Qwen2.5-VL-7B, Qwen2.5-VL-72B, and Qwen3-VL-32B as a representative case study, we present further analysis of the failure modes. We observed that the models **fail completely** on some tag combinations, which are listed in the Appendix A.5.
> >
> > In addition to these complete failures, we visualize the 10 combinations with the lowest $P_{50}$ in the Appendix (see Section A.5, Figure 9). The corresponding quantitative results in text are provided below.
> >
> > **Our Conclusion.** Although we *don't find exactly the same combinations* that appear in each model, we observe the absolute and relative position are prevalent in the majority of the failure modes. This suggests that _referring expressions involving spatial relationships pose a significant challenge_ for such models.
> >
> > **Table Q1-1.** LLaVA-Next-Mistral-7B:
> >
> > | Combinations                                                 | $P_{50}$ |
> > | ------------------------------------------------------------ | -------- |
> > | 3D Position + Possible Status + Relative Position            | 15.38    |
> > | 2D Position + Attribute + Possible Usage + Relative Position | 25.00    |
> > | Possible Status + Possible Usage + Reasoning                 | 25.00    |
> > | 2D Position + Attribute + Relative Position                  | 25.93    |
> > | 3D Position + Attribute + Relative Position                  | 29.67    |
> > | 3D Position + Relative Position + Size                       | 33.33    |
> > | Attribute + Reasoning + Relative Position                    | 33.33    |
> > | Reasoning + Size                                             | 33.33    |
> > | 3D Position + Reasoning + Relative Position                  | 33.33    |
> > | 2D Position + 3D Position                                    | 33.33    |
> >
> > **Table Q1-2.** Qwen2.5-VL-7B:
> >
> > | Combinations                                                                | $P_{50}$ |
> > | --------------------------------------------------------------------------- | -------- |
> > | 2D Position + Attribute + Possible Usage + Relative Position                | 25.00    |
> > | 3D Position + Attribute + Relative Position                                 | 31.58    |
> > | Attribute + Reasoning + Relative Position                                   | 33.33    |
> > | Interaction + Reasoning                                                     | 33.33    |
> > | 3D Position + Interaction + Possible Status + Relative Position             | 33.33    |
> > | 3D Position + Attribute + Interaction + Possible Status + Relative Position | 33.33    |
> > | Reasoning + Size                                                            | 33.33    |
> > | Attribute + Interaction + Possible Status                                   | 34.04    |
> > | 3D Position + Attribute                                                     | 37.50    |
> > | Interaction + Possible Status + Relative Position                           | 37.50    |
> >
> > **Table Q1-3.** Qwen2.5-VL-72B:
> >
> > | Combinations                                                    | $P_{50}$ |
> > | --------------------------------------------------------------- | -------- |
> > | 3D Position + Attribute + Possible Status + Relative Position   | 20.00    |
> > | 2D Position + Relative Position + Size                          | 33.33    |
> > | 3D Position + Relative Position + Size                          | 33.33    |
> > | Interaction + Reasoning                                         | 33.33    |
> > | 2D Position + Attribute + Possible Status + Relative Position   | 40.00    |
> > | 3D Position + Interaction + Possible Status + Relative Position | 44.44    |
> > | 2D Position + Possible Status                                   | 50.00    |
> > | Attribute + Possible Usage + Reasoning                          | 50.00    |
> > | 3D Position + Attribute                                         | 50.00    |
> > | 2D Position + Possible Status + Reasoning + Relative Position   | 50.00    |
> >
> > **Table Q1-4.** Qwen3-VL-32B:
> >
> > | Combinations                                                  | $P_{50}$ |
> > | ------------------------------------------------------------- | -------- |
> > | 3D Position + Possible Usage                                  | 33.33    |
> > | 2D Position + Relative Position + Size                        | 33.33    |
> > | Interaction + Reasoning                                       | 33.33    |
> > | Reasoning + Size                                              | 33.33    |
> > | 3D Position + Relative Position + Size                        | 33.33    |
> > | 3D Position + Attribute + Possible Status + Relative Position | 40.00    |
> > | Attribute + Possible Usage + Reasoning                        | 50.00    |
> > | 2D Position + 3D Position                                     | 50.00    |
> > | 2D Position + Attribute + Possible Usage + Relative Position  | 50.00    |
> > | Interaction + Possible Status + Size                          | 50.00    |

---

> > ### Author Response · Authors · 2025-11-27
> > **Q2. Examine Object-Specific Factors**
> >
> > > It would also be valuable to examine whether robustness to expression diversity correlates with object-specific factors, such as object size, category, or higher-level semantic grouping
> >
> > We evaluated the impact of three object-specific factors on Qwen2.5-VL-72B: (1) object size; (2) location distribution; and (3) object category. We have revised the paper and put the details in Appendix (Section A.6, Figure 10, 11, 12).
> >
> > - **Impact of Object Size**: Following COCO's [1] size classification criteria, we analyzed the hard pass rate across varying object sizes (see Section A.6, Figure 10). The results indicate a positive correlation between size and performance: the model achieves a higher hard pass rate on larger objects, with performance declining as object size decreases.
> > - **Impact of Object Location Distribution**: Similar to DETR [2], we normalized object center coordinates to a uniform scale (see Section A.6, Figure 11). The analysis demonstrates that the model's performance is largely robust to spatial variation, showing little correlation with the normalized object locations.
> > - **Object Class**: We visualized the hard pass rate across categories at a high semantic level (see Section A.5, Figure 12). We observed that performance variations across the majority of categories are minimal. We observe lower scores for specific semantic categories, such as "containers and storage". We attribute this to intrinsic visual properties common to these objects, such as transparency (e.g., glass cups) and small spatial dimensions, which are known challenges in this domain.
> >
> > [1] Lin, Tsung-Yi, Michael Maire, Serge Belongie, James Hays, Pietro Perona, Deva Ramanan, Piotr Dollár, and C. Lawrence Zitnick. "Microsoft coco: Common objects in context." In _European conference on computer vision_, pp. 740-755. Cham: Springer International Publishing, 2014.
> >
> > [2] Carion, Nicolas, Francisco Massa, Gabriel Synnaeve, Nicolas Usunier, Alexander Kirillov, and Sergey Zagoruyko. "End-to-end object detection with transformers." In _European conference on computer vision_, pp. 213-229. Cham: Springer International Publishing, 2020.

---

> > ### Author Response · Authors · 2025-11-27
> > **Q3. Regarding the Number of Objects per Image**
> >
> > > The manuscript states that the benchmark contains 824 objects, and the appendix notes 824 images, implying that each image contains exactly one annotated object. If this is indeed the intended dataset design, it would be helpful to emphasize this point clearly in the main text, as readers might otherwise question.
> >
> > Your understanding is correct. In our annotation process, we limit annotations to a single valid object per image. This strategy allows us to **prioritize scene diversity** within our annotation budget. We have clarified this design choice in **Section 3.2** (line 267, page 5) of the revised manuscript.

---

> > ### Author Response · Authors · 2025-11-27
> > **Q4. Discussion on Multiple Objects in a Scene**
> >
> > > Furthermore, the authors may wish to discuss how the evaluation framework generalizes to scenarios involving multiple target objects within the same scene, and whether current models show differing behavior when handling single-target versus multi-target referring expressions.
> >
> > Although our work aims to expose the robustness problem in REC task, we are willing to discuss the multi-target referring expression. We believe it is necessary to distinguish between two scenarios:
> >
> > - **Multiple objects annotated within a single scene, where each text query refers to a single object:** Our evaluation framework is designed to assess different descriptions of any given object. Since this process is independent of the number of annotated objects in a scene, we believe our evaluation framework can be directly extended to such scenarios.
> > - **Multiple objects are referred to by a single text query within a scene (e.g., HumanRef [1]):** This presents greater challenges for both dataset construction and model localization. Models may produce false positives (over-detection) and false negatives (missed detection). In such scenarios, we define a "hard pass" criterion where the model must accurately localize all targeted objects without omissions or false positives. _Specifically, a prediction is considered a "success" only if: (1) all targets are localized with an IoU > 0.5; and (2) there are no over-detections._ If we term this definition "multi-object success," it can directly replace the "success" in Equation (1) of our paper, resulting in $R^{\text{Hard}} = \frac{ 1 }{|\mathcal{O}|} \sum_{o^M}^{|\mathcal{O}|} \mathbf{1} [ (\sum_{e}^{|\mathcal{E}|} s_{o^M, e}) \ge |\mathcal{E}|]$, where $s_{o^M}$  means success for all targets referred in the expression $e$. Although this extension falls outside the scope of the current paper, we consider it an valuable research direction and added this as future work.
> >
> > We have put them in the revised manuscript (Section 5, page 10).
> >
> > [1] Jiang, Qing, Lin Wu, Zhaoyang Zeng, Tianhe Ren, Yuda Xiong, Yihao Chen, Liu Qin, and Lei Zhang. "Referring to any person." In _Proceedings of the IEEE/CVF International Conference on Computer Vision_, pp. 21667-21678. 2025.

---

### Official Review · Reviewer_xEkX · 2025-10-31

**Soundness:** 3
**Presentation:** 3
**Contribution:** 2
**Rating:** 4
**Confidence:** 4

**Summary:**

The paper presents DRef, a referring expression benchmark that evaluates models at the object level rather than per-expression. Each object is annotated with many positive and negative expressions, and the paper proposes strict metrics (Hard Pass Rate, Mean Consistency Rate) to test whether a model can localize the same object under diverse linguistic descriptions. Experiments on strong VLMs show that current models look good on single-expression REC but collapse when forced to be consistent across all expressions for an object.

**Strengths:**

1. The dataset is more varied than standard REC sets in that each object is associated with several types of referring signals (location, relative, attribute, interaction, negatives), so it can surface some failure modes that single-expression datasets miss.
2. The writing and organization are clear and easy to follow.

**Weaknesses:**

1. The experimental coverage is limited: the current comparisons do not include representative LLaVA series models, frontier models (e.g., GPT-4o, Gemini) , or other grounding models (e.g., GLAMM).
2. Prompt diversity is partly curator-driven and may be somewhat templated; it is unclear whether performance would hold under paraphrased, reordered, or more colloquial expressions.
3. While the paper diagnoses a gap in current REC evaluation, it does not propose a method aimed at improving the newly introduced metrics. The paper would be stronger with a proposed mitigation method to show that the proposed benchmark is not only harder but also improvable.
4. The benchmark can be viewed as a strong extension that emphasizes per-object multi-expression consistency, rather than a fundamentally new evaluation paradigm.

**Questions:**

1. Have you tested paraphrased, reordered, or more colloquial variants of the same expressions to see whether Hard Pass and consistency remain stable?
2. Since you have identified “same object, many expressions” as the key failure, have you tested any method that can actually improves Hard Pass?

---

> ### Author Response · Authors · 2025-11-27
>
> Thank you for your valuable feedback to help us improve our paper. We have revised our paper based on your feedback. We detail our response below and please kindly let us know if our response addresses your concerns.

---

> > ### Author Response · Authors · 2025-11-27
> > **Q1. Additional Frontier Models**
> >
> > > The experimental coverage is limited: the current comparisons do not include representative LLaVA series models, frontier models (e.g., GPT-4o, Gemini) , or other grounding models (e.g., GLAMM).
> >
> > In the Appendix, we originally provided results for additional models, such as the LLaVA series and Gemini-1.5-Flash. Here, we further provide results of GLAMM and other advanced models (Qwen3-VL series, Gemini-2.5-Pro, Gemini-3-Pro, OpenAI GPT-5, and OpenAI o4-mini) to highlight current limitations. Since GLAMM is designed to predict segmentation masks, we derive the corresponding bounding boxes by computing the minimum axis-aligned rectangle that encapsulates the predicted mask. We have revised the paper and put the results and discussion in Section 4.2 (Table 3 in page 8, line 427 page 8 to line 452 page 9). The full results of all models accessed are available in Appendix (A.1, Table 7).
> >
> > **Table Q1**: Grounding results of the frontier multi-modal models.
> >
> > | Models         | Pos $R^{\text{hard}}_{50}$ | Pos $P_{50}$ | Pos $R^{\text{soft}}_{50}$ | Pos $R^{C}$ | Neg $R^{\text{hard}}$ | Neg $P$ |
> > | -------------- | -------------------------- | ------------ | -------------------------- | ----------- | -------------------------- | ------------ |
> > | GLAMM (Bbox)   | 6.7                        | 47.9         | 74.8                       | 28.6        | 0.0                        | 0.3          |
> > | GLAMM (Mask)   | 9.0                        | 50.1         | 78.5                       | 27.7        | 0.0                        | 0.3          |
> > | Gemini-2.5-Pro | 3.2                        | 6.6          | 9.6                        | 12.3        | 1.3                        | 26.7         |
> > | Gemini-3-Pro   | 3.3                        | 6.7          | 7.8                        | 12.8        | 0.4                        | 21.3         |
> > | OpenAI GPT-5   | 0.7                        | 9.2          | 30.0                       | 9.0         | 1.1                        | 24.8         |
> > | OpenAI o4-mini | 2.5                        | 27.1         | 56.8                       | 13.3        | 6.8                        | 53.0         |
> > | Qwen3-VL-2B    | 17.2                       | 51.8         | 80.3                       | 32.5        | 0.0                        | 0.0          |
> > | Qwen3-VL-4B    | 19.5                       | 67.8         | 94.2                       | 41.9        | 0.0                        | 3.2          |
> > | Qwen3-VL-8B    | 18.8                       | 67.5         | 95.0                       | 40.9        | 0.0                        | 1.1          |
> > | Qwen3-VL-32B   | 32.8                       | 78.1         | 97.5                       | 53.8        | 0.5                        | 21.1         |
> >
> > We have relocated the results of the Gemini and LLaVA series from the Appendix to the main text and incorporated the results of GLAMM, Qwen3-VL, Gemini-3-Pro, GPT-5, and o4-mini to improve the presentation of our experimental section. Additionally, we have also included a discussion on the results of these closed-source models.
> >
> > **Discussion on the results of advanced closed-source multi-modal models.** We observe that GPT-5, o4-mini, Gemini-2.5-Pro and Gemini-3-Pro yield poor results. We suspect that these frontier closed-source models have not been specifically optimized for grounding tasks. To the best of our knowledge, **there is almost no existing literature directly employing** these frontier models for grounding tasks. Furthermore, neither the technical report of Gemini-2.5-Pro [1] nor the model cards of GPT-5 and o4-mini [3,4] include evaluations on visual grounding tasks to demonstrate their capabilities in this domain. While the Qwen2.5-VL technical report [2] includes benchmarks using Gemini-1.5-Pro, the results indicate that Gemini-1.5-pro significantly underperforms compared to other models on grounding tasks, which serves as further evidence that these frontier models lack specific optimization for visual grounding.
> >
> > [1] Comanici, Gheorghe, Eric Bieber, Mike Schaekermann, Ice Pasupat, Noveen Sachdeva, Inderjit Dhillon, Marcel Blistein et al. "Gemini 2.5: Pushing the frontier with advanced reasoning, multimodality, long context, and next generation agentic capabilities." _arXiv preprint arXiv:2507.06261_ (2025).
> >
> > [2] Bai, Shuai, Keqin Chen, Xuejing Liu, Jialin Wang, Wenbin Ge, Sibo Song, Kai Dang et al. "Qwen2. 5-vl technical report." _arXiv preprint arXiv:2502.13923_ (2025).
> >
> > [3] https://openai.com/index/introducing-gpt-5/
> >
> > [4] https://openai.com/index/introducing-o3-and-o4-mini/

---

> > ### Author Response · Authors · 2025-11-27
> > **Q2. Rewriting Referring Expressions**
> >
> > > Prompt diversity is partly curator-driven and may be somewhat templated; it is unclear whether performance would hold under paraphrased, reordered, or more colloquial expressions. Have you tested paraphrased, reordered, or more colloquial variants of the same expressions to see whether Hard Pass and consistency remain stable?
> >
> > We implemented two rewriting strategies on the existing texts: (1) colloquial adaptation and (2) modifier reordering. We have revised the paper and put the results and discussion in Section 4.2 (line 511 to 526 and Table 6, page 10)
> >
> > **Our Conclusion:** Our evaluation results indicate that _the primary challenge remains the inherent diversity in description perspectives_, though these rewriting strategies introduce minor perturbations. Our original referring expressions emphasize the diversity in description perspectives, capturing how a single object within a scene can be targeted via distinct textual descriptions. Although rewriting the texts with different linguistic style presented a challenge to the evaluated models, it did not result in significant performance degradation in hard pass rate and mean consistency rate.
> >
> > This finding provides valuable insight, clarifying that limitations in model grounding robustness stem primarily from the _descriptive perspective_ (semantic content) of visual elements, rather than stylistic factors such as colloquialism or syntactic ordering. We have incorporated this analysis into Section 4.2.
> >
> > **Experimental Details:**
> > - **Referring Expression Rewriting:** We utilized GPT-5 to rewrite the existing texts. We carefully designed the prompts to ensure the model preserved the semantics of the original referring expressions, with specific instructions to retain the original text if confidence in the rewriting was low.
> > - **Evaluation Results**: We evaluated the Qwen2.5-VL-7B model. The results for the different rewriting strategies are presented in the table below:
> >
> > **Table Q2.** Evaluation results of Qwen2.5-VL-7B on different linguistic style.
> >
> > | Text Style         | Pos $R^{\text{hard}}_{50}$ | Pos $P_{50}$ | Pos $R^{\text{soft}}_{50}$ | Pos $R^{C}$ | Neg $R^{\text{hard}}$ | Neg $P$ |
> > | ------------------ | -------------------------- | ------------ | -------------------------- | ----------- | -------------------------- | ------------ |
> > | Originally Labeled | 18.4                       | 57.8         | 88.0                       | 35.7        | 0.0                        | 0.0          |
> > | Colloquial         | 17.5                       | 60.0         | 90.3                       | 36.8        | 0.0                        | 0.0          |
> > | Reordered          | 17.2                       | 59.3         | 90.3                       | 36.2        | 0.0                        | 0.0          |

---

> > ### Author Response · Authors · 2025-11-27
> > **Q3. Potential Solution**
> >
> > > While the paper diagnoses a gap in current REC evaluation, it does not propose a method aimed at improving the newly introduced metrics. The paper would be stronger with a proposed mitigation method to show that the proposed benchmark is not only harder but also improvable.
> >
> > **Our Conclusion:** Our preliminary experiments suggest that **Reinforcement Fine-Tuning (RFT)** is a promising direction for improving the robustness of REC models.
> >
> > **Our Solution:** We conducted a two-stage fine-tuning process on Qwen2.5-VL-7B using 368 **additionally labeled objects** in extra images. Specifically, we applied Supervised Fine-Tuning (SFT) followed by RFT with GRPO on the same training data. We denote this model as Qwen2.5-VL-7B-SFT-RFT. We have revised the paper and put the details and results to Section 4.1 (line 375 to 377, page 7) and Section 4.2 (Table 3, page 8; line 427 page 8 to 451 page 9).
> >
> > **Table Q3.** Results of different fine-tuning policy for Qwen2.5-VL-7B.
> >
> > | Models                   | $R^{\text{hard}}_{50}$ | $P_{50}$ | $R^{\text{soft}}_{50}$ | $R^{C}$ |
> > | ------------------------ | ---------------------- | -------- | ---------------------- | ------- |
> > | Qwen2.5-VL-7B (baseline) | 18.4                   | 57.8     | 88.0                   | 35.7    |
> > | Qwen2.5-VL-7B-SFT        | 16.9                   | 66.3     | 94.2                   | 39.8    |
> > | $\Delta$                 | -1.5                   | +8.5     | +6.2                   | +4.1    |
> > | Qwen2.5-VL-7B-RFT        | 24.5                   | 74.7     | 95.8                   | 50.5    |
> > | $\Delta$                 | +6.1                   | +16.9    | +7.8                   | +14.8   |
> > | Qwen2.5-VL-7B-SFT-RFT    | 30.0                   | 74.7     | 94.1                   | 52.2    |
> > | $\Delta$                 | +11.6                  | +16.9    | +6.1                   | +16.5   |
> >
> > **Analysis:** Our Qwen2.5-VL-7B-SFT-RFT model achieves 30.0 on $R_{50}^{hard}$, significantly outperforming the baseline Qwen2.5-VL-7B (18.4) and even surpassing the much larger Qwen2.5-VL-72B (27.7). While the SFT stage enhances $P_{50}$​ (+8.5) and $R^C$(+4.1), it leads to a decline in $R_{50}^{\text{hard}}$(-1.5). This indicates a trade-off where the model learns something new but experiences some forgetting on challenging samples. Directly applying RFT with GRPO improves $P_{50}$​ (+16.9), $R^C$ (+14.8), and $R_{50}^{\text{hard}}$ (+6.1), showing it can improve the robustness of the model. Finally, adopting the two-stage fine-tuning process further improves the baseline on $P_{50}$​ (+16.9), $R^C$ (+16.5), and $R_{50}^{\text{hard}}$ (+11.6), indicating the initial SFT stage is important. We think SFT helps model learning to capture more details of an object.
> >
> > In a word, these results indicate that RFT is a potential solution for enhancing the robustness of VLMs against diverse referring expressions. Furthermore, as current RFT research predominantly targets domains such as mathematics and coding, we believe the underlying mechanisms and key factors of RFT in the context of robust visual grounding require further investigation.

---

> ### Author Response · Authors · 2025-11-27
> **Q4. Strong Extension to Existing Benchmarks**
>
> > The benchmark can be viewed as a strong extension that emphasizes per-object multi-expression consistency, rather than a fundamentally new evaluation paradigm.
>
> We thank the reviewers for recognizing our DRef benchmark as a strong complement to existing grounding benchmarks. To be clear, we didn't claim DRef is a new evaluation paradigm. We believe DRef will facilitate the development of more robust and reliable visual grounding models, thereby advancing downstream applications in domains such as Embodied AI and VLA.

---

### Author Response · Authors · 2025-11-27
**Global Response**

We would like to thank all reviewers for their time and valuable feedback, which have helped us significantly improve our manuscript. We are encouraged that Reviewers xEkX and 1Wzo agree that DRef effectively surfaces failure modes that single-expression datasets often miss. Reviewer 9YEC highlights that DRef is of high quality and complements current REC benchmarks well, and our proposed metrics effectively quantify VLM robustness and offer an interesting new assessment of model capacity. Finally, Reviewer TExx found our dataset to be well-motivated and the proposed metrics insightful.

We have addressed the specific concerns below and revised our manuscript accordingly.

---

### Author Response · Authors · 2025-12-03
**Summary**

We first thank the time and valuable feedback from all reviewers, ACs, SACs, and PCs, which have helped us improve our paper. We have addressed the concerns from four reviewers and revised our manuscript accordingly.

---

> ### Author Response · Authors · 2025-12-03
> **Highlights**
>
> We are encouraged by the reviewers’ recognition of our paper’s strengths:
>
> 1. **Consensus on Contribution:** All reviewers agree that our **DRef benchmark effectively exposes the limitations of existing REC benchmarks.**
> 2. **Data Quality & Diversity:** Reviewers xEkX, 9YEC, and TExx commended the **high diversity** of DRef. Furthermore, Reviewers 9YEC and 1Wzo recognized the **high quality** of the benchmark, specifically praising our rigorous two-pass annotation protocol for quality control.
> 3. **Novel Metrics:** Reviewers xEkX, 9YEC, and TExx considered our proposed metrics to be **insightful and interesting** (9YEC). They noted that these metrics:
>     - Quantify the **robustness** (9YEC, 1Wzo, TExx) and **consistency** (TExx) of VLMs.
>     - Expose failure modes missed by single-expression benchmarks (1Wzo) and reveal the limitations of current VLMs (TExx).
> 4. **Experimental Rigor:** Our evaluation was praised for being **comprehensive** (9YEC) as well as **rigorous and realistic** (TExx).
> 5. **Presentation:** Finally, reviewers appreciated the presentation of the work, noting that the **writing and organization are clear** (xEkX) and the **motivation is well-argued** (TExx).

---

> ### Author Response · Authors · 2025-12-03
> **Rebuttal Highlights**
>
> We thank the reviewers for their constructive comments and time. During the rebuttal phase, we have addressed the concerns raised. We summarize the key updates and responses below:
>
> - **Evaluation of Additional Models (Reviewer xEkX and TExx)**. Some suggested models are already in the appendix of our original manuscript. We moved them to Table 3 in the paper. We also include evaluation results of GLAMM, GPT-5, o4-mini, Gemini-2.5-Pro, Gemini-3-Pro, and Qwen3-VL series in Table 3 of the revised paper.
> - **Potential Solution (Reviewer xEkX and 1Wzo)**. We present a two-stage training pipeline comprising a Supervised Fine-Tuning (SFT) stage followed by a Reinforcement Fine-Tuning (RFT) stage. Our analysis demonstrates that RFT is a promising direction for improving robustness, though prior SFT is recommended to further improve performance. Details, results and the accompanying analysis have been added to the revised manuscript (Section 4.1 & 4.2).
> - **Dataset Scale (Reviewer 1Wzo and TExx)**. We demonstrate that the scale of annotated objects in our benchmark is **comparable** to other challenging multi-modal benchmarks. Furthermore, statistical analysis and experimental results show that while our annotated texts are comparable in volume to RefCOCO evaluation subsets, they present a significantly **higher challenge** due to the diversity of descriptions.
> - **Rewriting vs. Intrinsic Diversity (Reviewer xEkX)**. We clarify that the influence of rewriting styles is not the primary bottleneck for current models. Rather, the main challenge stems from the intrinsic diversity of describing aspects (e.g., attributes vs. spatial relationships), which is the core contribution of our benchmark. This analysis has been detailed in the revised paper (Section 4.2).
> - **Analysis of Failure Modes (Reviewer 9YEC)**. Our statistical breakdown reveals that current models most commonly fail on absolute and relative spatial positioning. Notably, we observed no identical combinations of failure tags across different models, suggesting diverse error patterns. This failure case analysis is now included in the appendix (Section A.5).
> - **Object Factor Analysis (Reviewer 9YEC)**. Our analysis indicates that model robustness is not significantly correlated with object location within the image, and variations across different semantic groups are minimal. However, we observed that models tend to be more robust when detecting larger objects. These findings are presented in the appendix (Section A.6).
> - **Quantitative Analysis of Perturbations (Reviewer TExx)**. While our initial submission presented qualitative analysis on why Qwen2.5-VL-3B appeared less perturbed than larger variants,, we benefit from the comment and present more quantitative analysis. Our analysis show that Qwen2.5-VL-3B prone to ignore the input text and output the same result, while the larger models are more responsive to specific details in the referring expressions. We present this analysis together with original qualitative analysis in the appendix (Section A.4).
> - **Number of Objects per Image (Reviewer 9YEC)**. We clarify that only one object is labeled per image to prioritize the scene diversity. We have revised the paper to avoid any potential misleading (Section 3.2).
> - **Discussion on Multiple Objects Extension (Reviewer 9YEC)**. While our benchmark focus on exposing the robustness problem with multiple referring expressions for one object, we agree that extending this to _multiple expressions for multiple objects_ is a valuable direction. We have revised Section 5 to discuss this multi-target scenario as future work.
> - **More Detailed Discussion with Related Work (Reviewer TExx)**. While we presented a simple discussion with related works, e.g., MMR and FineCops-Ref, in our initial submission, we follow the suggestion and expand the discuss with more details. We clarify that they are not explicitly designed to be multiple expressions for one object. The MMR is designed to be one expression for multiple objects, FineCops-Ref is designed to assign each expression to one of multiple difficulty levels. SCALAR-VG is designed for multiple tasks, including bounding boxes, keypoints, and polygons. We put this discussion to appendix (Section A.3).

---

> ### Author Response · Authors · 2025-12-03
> **Closing**
>
> We once again thank all reviewers for their thoughtful input. We are encouraged that the reviewers recognize DRef’s value in exposing the limitations of existing REC benchmarks. We believe the revised manuscript effectively addresses the remaining concerns and hope for your favorable consideration.

---

### Meta-Review · Area_Chair_AXnZ · 2026-01-08

**Summary:**

The paper received the mixed ratings. All the reviewers agrees that the DRef exposes the limitations of existing REC benchmarks for limited diversity of referring expressions for same objects and provides more various referring expressions than standard REC datasets. Furthermore, Reviewer 9YEC, TExx, 1Wzo recognize the proposed metrics (Hard Pass Rate and Mean Consistency Rate) are complementary and insightful. The authors have provided a detailed and thorough rebuttal in adding a wide range baselines   for evaluation and analysis (e.g., impact of object size, object class, and the distribution of object location, etc) to address the concerns of all the reviewers. Following the rebuttal, I conducted follow-up email communications with all reviewers to check their final assessments. Reviewer 1Wzo and xEkX give their final evaluation. Reviewer 1Wzo still think the dataset scale is relatively small which is also pointed out by Reviewer TExx initially. On the other hand,  Reviewers xEkX  appreciate the authors to address many of his concerns. However,  Reviewer xEkX thinks that the manuscript still requires further polishing, particularly to provide more detailed analysis and insight into the SFT/RFT beyond the improved empirical results presented in the rebuttal. Finally, Reviewer 1Wzo and xEkX both keep the same ratings of 4. Considering the remaining unresolved concerns, the paper is not recommended for acceptance in its current form. The authors are encouraged to further polish the manuscript by incorporating the reviewers’ suggestions and fully integrating the additional results developed during the discussion period for submission to a future venue.

**Reviewer Concerns:**

The authors provided a comprehensive rebuttal and addressed many concerns raised by the reviewers. For the final assessments, I followed up with all reviewers through emails. Reviewers xEkX and 1Wzo responded before the meta-review deadline and acknowledged the authors’ efforts in adding additional baseline comparisons and introducing a preliminary results of two-stage fine-tuning pipeline (SFT/RFT) to demonstrate the effectiveness of the benchmark. However, some concerns remain unresolved. Reviewer xEkX thinks that the manuscript still requires further polishing, particularly to provide more detailed analysis and insight into the SFT/RFT beyond the improved empirical results presented in the rebuttal (i.e.,  As mentioned by Reviewer xEkX, it does not clearly derive training objectives, error taxonomies, or targeted interventions from the specific failure patterns surfaced by DRef.). Reviewer 1Wzo still think the dataset as relatively small. As a result, both reviewers decided to maintain their original ratings with 4 (borderline rejects).

**Reviewer Scores:**

The authors provided a thorough and detailed rebuttal that addressed many of the concerns raised by the reviewers. I believe that some reviewers who initially gave negative ratings might have revised their scores if they had been able to participate full in the discussion and have more back-and-forth discussions and paper revision. However, as mentioned in reviewer concerns, Reviewers xEkX and 1Wzo still have unresolved concerns and would like the authors to further polish the manuscript.

---

### Decision · Program_Chairs · 2026-01-26

Reject